# When Is Generalizable Reinforcement Learning Tractable?

**Dhruv Malik**
Machine Learning Department
Carnegie Mellon University
Pittsburgh, PA 15213

**Yuanzhi Li**
Machine Learning Department
Carnegie Mellon University
Pittsburgh, PA 15213

**Pradeep Ravikumar**
Machine Learning Department
Carnegie Mellon University
Pittsburgh, PA 15213

## Abstract

Agents trained by reinforcement learning (RL) often fail to generalize beyond the environment they were trained in, even when presented with new scenarios that seem similar to the training environment. We study the query complexity required to train RL agents that generalize to multiple environments. Intuitively, tractable generalization is only possible when the environments are similar or close in some sense. To capture this, we introduce *Weak Proximity*, a natural structural condition that requires the environments to have highly similar transition and reward functions and share a policy providing optimal value. Despite such shared structure, we prove that tractable generalization is impossible in the worst case. This holds even when each individual environment can be efficiently solved to obtain an optimal linear policy, and when the agent possesses a generative model. Our lower bound applies to the more complex task of representation learning for efficient generalization to multiple environments. On the positive side, we introduce *Strong Proximity*, a strengthened condition which we prove is sufficient for efficient generalization.

## 1  Introduction

Reinforcement learning (RL) is the dominant paradigm for sequential decision making in machine learning, and has achieved success in a variety of domains such as competitive gaming [33, 40] and robotic control [21, 22]. Despite this success, many issues prevent RL from being regularly used in the real world. For example, one typically trains and tests RL agents in the same environment. In such cases, an agent can memorize behavior that achieves high reward, without acquiring the true behavior that the system designer desires. This has raised concerns about RL agents overfitting to a single environment, instead of learning meaningful skills [17].

Indeed, a long line of work has noted the brittleness of RL agents: slight changes in the environment, such as those incurred by modeling or simulator design errors, or slight perturbations of the agent's trajectory, can lead to catastrophic declines in performance [36, 49, 23]. Furthermore, although RL agents can solve difficult tasks, they struggle to transfer the skills they learned in one task to perform well in a different but similar task [37, 48]. Yet, in the real world, it is reasonable to expect that RL agents will see scenarios that are at least mildly different from the specific scenarios they trained for.

Hence, a desirable property of RL agents is that of *generalization*, broadly defined as the ability to discern the correct notion of behavior and perform well in semantically similar environments. We focus on two popular generalization settings. The *Average Performance* setting assumes there is an

35th Conference on Neural Information Processing Systems (NeurIPS 2021).

underlying distribution over the environments that an agent might encounter. The agent's goal is to perform well on average across this distribution [35, 34, 11]. The *Meta Reinforcement Learning* setting is closely related [20, 10, 37]. Here an agent first learns from a suite of training environments sampled from a distribution. Then at test time the agent must leverage this experience to adapt to a new environment sampled from the same distribution, via only a few queries in the new environment.

Of course, in full generality, both notions of generalization are impossible to achieve efficiently. This is especially true in the RL function approximation setting, where the *cardinality* of the state space is potentially infinite, and so we desire query complexity that scales (polynomially) with the *dimensionality* of the state space [13, 38, 14, 31, 46]. Hence, key to both lines of inquiry is the premise that the environments are structurally similar. For example, a robot may face the differing tasks of screwing a bottle cap and turning a doorknob, but both tasks involve turning the wrist [37]. The hope is that if the environments are sufficiently similar, then RL can exploit this structure to efficiently discover policies that generalize.

Yet, it remains unclear what kind of structure is necessary, and what it means for different environments to be close or similar. Motivated by this, we ask the following question:

**What are the structural conditions on the environments that permit efficient generalization?**

This question underlies the analysis of our paper. We focus on environments that share state-action spaces, since even this basic case is not well understood in the literature. Indeed, even in this simplified setting, efficient generalization can be highly non-trivial. We make the following contributions.

**Our Contributions.** We introduce *Weak Proximity*, a natural structural condition that is motivated by classical RL results, and requires the environments to have highly similar transition and reward functions and share optimal trajectories. We prove a statistical lower bound demonstrating that tractable generalization is impossible, despite this shared structure. This lower bound holds even when each individual environment can be efficiently solved to obtain an optimal linear policy, and when the agent possesses a generative model. Consequentially, we show that a classical metric for measuring the relative closeness of MDPs is not the right metric for modern RL generalization settings. Our lower bound implies that learning a state representation for the purpose of efficiently generalizing to multiple environments, is worst case sample inefficient — even when such a representation exists, the environments are ostensibly similar, and any single environment can be efficiently solved.

To provide a sufficient condition for efficient generalization, we introduce *Strong Proximity*. This structural condition strengthens Weak Proximity by additionally constraining the environments to share an optimal policy. We provide an algorithm which exploits Strong Proximity to provably and efficiently generalize, when the environments share deterministic transitions.

## 2   Related Work

**Simulation Lemma.** Many prior works define notions of statistical distance between Markov decision processes (MDPs), and measure the relative value of policies when deployed in different MDPs that are close under such metrics. The Simulation Lemma, which uses total variation distance between transitions and the absolute difference of rewards as this metric, is a well known formalization of this and has been very useful in classical prior work [26, 27, 6, 25, 1]. These works do not directly tackle generalization, but their analyses construct an approximate MDP that models the true MDP under the aforementioned metric. Solving this approximate MDP then corresponds to solving the true MDP. It is natural to ask whether this metric is useful for measuring the similarity of MDPs in modern RL generalization settings. We show this metric is not appropriate for the settings we study.

**Transfer & Multitask Learning.** There are varying formalisms of both settings, so we do not directly study them. However, they are broadly relevant, and we expect our theory to be useful for future studies of these settings. The works [32, 7, 24, 43] all study metrics for measuring variation between MDPs that are different from the metrics we study. A metric similar to the one used in the Simulation Lemma has also been studied [18], and we show that this is inappropriate for our settings.

**Average Performance & Meta RL Settings.** We directly study these two settings, which have seen much empirical work [35, 11, 12, 37, 48]. On the theoretical side, [5, 42] study an Average Performance setting where the agent receives a noisy observation in lieu of the actual state. We focus on the simpler setting where the agent knows its state. Recent works [16, 44] analyze the

MAML algorithm [20] in the context of Meta RL. In the worst case, their complexity bounds scale exponentially with the horizon, and they do not discuss structure which permits tractable Meta RL.

**Representation Learning.** A large body of work has focused on extracting a representation useful for a single MDP [19, 8, 30, 50]. Some works extend this to multiple MDPs [9, 3, 41], but they are about learning shared representations for MDPs that appear similar (but not from a sample efficiency perspective), while we formalize what it means for MDPs to be similar (in a sample efficient sense). Indeed, these works study the general case when the environments have distinct state spaces, but our lower bounds show generalization is non-trivial even when each MDP shares the same state space.

## 3 Problem Formulation

**Notation & Preliminaries.** Before describing our settings of interest, we establish notation and briefly review preliminaries. We always use $M$ to denote a Markov decision process (MDP). Recall that an undiscounted finite horizon MDP is specified by a set of states $\mathcal{S}$, a set of actions $\mathcal{A}$, a transition function $\mathcal{T}$ which maps from state-action pairs to distributions over states, a reward function $R$ which maps state-action pairs to nonnegative real numbers, and a finite planning horizon $H$. We assume that the state-action pairs are featurized, so that $\mathcal{S} \times \mathcal{A} \subset \mathbb{R}^d$, and that $\|(s,a)\|_2 = 1$ for all $(s,a) \in \mathcal{S} \times \mathcal{A}$. Any MDP we consider is undiscounted and has a finite action space, but could have an uncountable state space. If we need to refer to the transition or reward function of a specific MDP $M$, then we shall denote this via $\mathcal{T}_M$ or $R_M$. We will denote a distribution over MDPs as $\mathcal{D}$. We also assume that $\mathcal{S}$ can be partitioned into $H$ different levels. This means that for each $s \in \mathcal{S}$ there exists a unique $h \in \{0, 1 \ldots H-1\}$ such that it takes $h$ timesteps to arrive at $s$ from $s_0$. We say that such a state $s$ lies on level $h$, and denote $\mathcal{S}_h$ to be the set of states on level $h$. This assumption is without loss of generality, since we can always make the final coordinate of each state-action pair encode the number of timesteps that elapsed to reach the state. A "deterministic MDP" is one with deterministic transitions. For any MDP, we assume a single initial state $s_0$, which strengthens our lower bounds.

A policy maps each state to a corresponding distribution over actions, and shall typically be denoted by $\pi$. The total expected reward accumulated by policy $\pi$ when initialized at state $s$ in MDP $M$ is given by $\mathbb{E}\left[\sum_{h=\text{level}(s)}^{H-1} R_M(s_h, a_h) \mid \pi\right]$ and will be denoted by $V_M^s(\pi)$. Here the expectation is over the trajectory $\{(s_h, a_h)\}_{h=\text{level}(s)}^{H-1}$ given that the first state in the trajectory is $s$. So $V_M^s(\pi)$ is the value of the policy $\pi$ in MDP $M$ with respect to (w.r.t) initial state $s$. Analogously, if a policy is parameterized by $\overline{\theta} = \{\theta_h\}_{h=0}^{H-1}$, then we denote it as $\pi(\overline{\theta})$, and the notation $V_M^s(\pi)$ is then replaced by $V_M^s(\overline{\theta})$. We assume that the cumulative reward collected by any trajectory from any initial state $s$ in any MDP $M$ is always bounded by 1. Hence the value of any policy in any MDP lies in the interval $[0, 1]$. $\mathbb{TV}(P, Q)$ denotes the total variation (TV) distance between probability distributions $P$ and $Q$.

### 3.1 Problem Settings

**Average Performance Setting.** There is a fixed distribution $\mathcal{D}$ over a family of MDPs. One can sample MDPs from $\mathcal{D}$. The algorithm can query states in the sampled MDPs, to learn some common structure. The goal is to solve

$$\max_{\pi} \mathbb{E}_{M \sim \mathcal{D}}\left[V_M^{s_0}(\pi)\right]. \tag{1}$$

**Meta Reinforcement Learning Setting.** There is a fixed distribution $\mathcal{D}$ over a family of MDPs. At training time, one can sample MDPs from $\mathcal{D}$. The algorithm can query states in the sampled MDPs, to learn some common structure between all the MDPs. Then at test time, an MDP $M_{\text{test}}$ is sampled from the same distribution $\mathcal{D}$. The goal of the algorithm is to learn a subroutine, which with non-trivial probability over the selection of $M_{\text{test}}$, can solve

$$\max_{\pi} V_{M_{\text{test}}}^{s_0}(\pi), \tag{2}$$

significantly more efficiently than trying to solve $M_{\text{test}}$ without having seen any training MDPs.

In both settings, "sampling an MDP" means drawing an MDP i.i.d from $\mathcal{D}$, so that the agent can then interact with it by performing trajectories in it. Note that in Eqs. (1) & (2), in full generality the initial state $s_0$ is random and depends on $M, M_{\text{test}}$. We focus on the case when the MDPs supporting $\mathcal{D}$ share a state-action space, and hence share the same single initial $s_0$ since we assume a single

initial state for any MDP. While such assumptions are already strong, they only strengthen our lower bounds. Furthermore, it is necessary to understand this simpler setting, before looking at more complex scenarios. To the best of our knowledge, such a study has not appeared in prior work.

To solve the problems described by Eqs. (1) & (2), we need to define an appropriate query model for the algorithm. We consider two query models, the first of which is strictly stronger than the second.

**Strong Query Model (SQM).** Sampling an MDP from $\mathcal{D}$ incurs no cost. The agent has a generative model of any sampled MDP $M$. To interact with $M$, the agent inputs a state-action pair $(s, a)$ of $M$ into the model, and receives $R_M(s, a)$ and a state sampled from $\mathcal{T}_M(s, a)$. This incurs a query cost of one. The goal is to solve Eqs. (1) & (2) with total query cost that is at most polynomial in $|\mathcal{A}|, H, d$.

**Weak Query Model (WQM).** Sampling an MDP from $\mathcal{D}$ incurs a query cost of $q_{\mathcal{D}} \geq 1$. Within a sampled MDP $M$, the agent operates in the standard episodic RL setup. Concretely, during each episode the agent interacts with the MDP by starting from $s_0$, taking an action and observing the next state and reward, and repeating. Each action taken during an episode incurs a query cost of one. The goal is to solve Eqs. (1) & (2) with total query cost that is at most polynomial in $q_{\mathcal{D}}, |\mathcal{A}|, H, d$.

Note that under both SQM and WQM, we desire query cost that is polynomial in the dimension $d$ of the state-action space, as opposed to the cardinality of the state space. This is standard for our function approximation setting [13, 38, 14, 31, 46], since the cardinality of the state space could be infinite. Also, we separate SQM and WQM because it is well known that different query models can lead to various subtleties in analysis and sample complexity guarantees [13, 38, 14, 31, 46]. The generative model that defines SQM assumes that we can simulate any state of our choice without performing a trajectory, which is unrealistic in practice, and is one of the strongest oracle models considered in prior literature [28, 4, 39, 14, 31, 2]. We shall present our lower bounds under SQM, which makes these results stronger, but shall present our upper bound under the natural and standard WQM.

Without any conditions on $\mathcal{D}$, the Average Performance & Meta RL settings can be intractable, even under SQM. This will occur if the MDPs supporting $\mathcal{D}$ do not share structure. This will also occur if any individual MDP cannot be solved efficiently. Nevertheless, in practice one often deals with MDPs which share meaningful structure [11, 37]. For instance, the transition distributions of the MDPs may be close in a suitable metric. Similarly, the reward functions of the MDPs might be close in an appropriate norm, or each MDP may share a set of optimal trajectories. And in practice, individual MDPs can usually be optimized efficiently [35, 48]. In such cases, it is reasonable to expect tractable generalization. We are interested in formalizing conditions that permit efficient generalization. We will particularly focus on conditions which capture shared structure of the MDPs and the tractability of individual MDPs. We now formally state the problem we consider throughout our paper.

*Which conditions on $\mathcal{D}$ allow us to solve the Average Performance & Meta RL settings efficiently?*

As mentioned above, there are two types of requirements. The first requirement should ensure that the MDPs are meaningfully similar. We formalize such conditions in Section 3.2. The second requirement should ensure that any individual MDP is efficiently solvable, else there is no hope to efficiently find policies that generalize for many MDPs. We formalize such properties in Section 3.3.

### 3.2 Strong & Weak Proximity

We now identify conditions that capture when the MDPs supporting $\mathcal{D}$ share meaningful structure. Since MDPs are defined in terms of rewards and transitions, it is very natural to impose conditions directly on the rewards and transitions. To this end, we state the following condition.

**Condition 1 (Similar Rewards & Transitions)** *The distribution $\mathcal{D}$ satisfies this condition with parameters $\xi_{\mathrm{r}}, \xi_{\mathrm{tr}} \geq 0$ when:*

*(a) Each MDP supporting $\mathcal{D}$ shares the same state-action space $\mathcal{S} \times \mathcal{A}$.*

*(b) For all $M_i, M_j$ supporting $\mathcal{D}$ and all $(s, a) \in \mathcal{S} \times \mathcal{A}$ we have $|R_{M_i}(s, a) - R_{M_j}(s, a)| \leq \xi_{\mathrm{r}}$.*

*(c) For all $M_i, M_j$ supporting $\mathcal{D}$ and all $(s, a) \in \mathcal{S} \times \mathcal{A}$ we have $\mathbb{TV}(\mathcal{T}_{M_i}(s, a), \mathcal{T}_{M_j}(s, a)) \leq \xi_{\mathrm{tr}}$.*

The parameters $\xi_r, \xi_{tr}$ naturally quantify the similarity of different MDPs. Conditions of this form are canonical and have yielded fruitful research in classical RL literature [26, 27, 6, 25, 1], in the guise of the Simulation Lemma (see Section 2). To concretize this condition with an example, consider a suite of simulated robotic goal reaching tasks [48], where the physics simulator is the same in each task, so the transitions are fixed and $\xi_{tr} = 0$, but the goal location changes from task to task, implying that $\xi_r > 0$. We now establish our Weak Proximity condition, which strictly strengthens Condition 1.

**Condition 2 (Weak Proximity)** *The distribution $\mathcal{D}$ satisfies Weak Proximity with parameters $\xi_r, \xi_{tr}, \alpha \geq 0$ when:*

(a) *$\mathcal{D}$ satisfies Condition 1 with parameters $\xi_r, \xi_{tr} \geq 0$.*

(b) *There exists a deterministic policy $\pi^\star$ which for any MDP $M$ satisfies $V_M^{s_0}(\pi^\star) \geq \max_{\pi'} V_M^{s_0}(\pi') - \alpha$.*

Weak Proximity strengthens Condition 1 by additionally requiring (via part (b)) that there exists some policy $\pi^\star$ which provides $\alpha$-suboptimal value for each MDP supporting $\mathcal{D}$. Intuitively, this condition implicitly constrains the MDPs to be similar, since there is a single policy which provides (nearly) optimal value, irrespective of the MDP it is deployed in. Furthermore, recall from Eqs. (1) & (2) that the objectives of the Average Performance & Meta RL settings are defined in terms of value w.r.t the initial state $s_0$. So it is natural to assume, as we do in part (b), that there is one policy which provides good value w.r.t $s_0$ for all MDPs. From an algorithmic perspective, this is helpful, because it ensures that we can restrict our search to those policies which perform well for many MDPs supporting $\mathcal{D}$.

To concretize this condition in our aforementioned example of simulated robotic goal reaching tasks [48], consider a suite of tasks where each task has multiple different equivalent goals (so the task is complete when the robot reaches any single one of these goals), but there is only one goal location that is shared and invariant across each task. The trajectory that leads to this goal location from $s_0$ defines a policy $\pi^\star$, such that for any task $M$ we have $V_M^{s_0}(\pi^\star) = \max_{\pi'} V_M^{s_0}(\pi')$, implying that Weak Proximity is satisfied with $\alpha = 0$.

Although Condition 1 is natural and well motivated by classical RL literature, it (and Weak Proximity) may seem strong. This is because it requires that each MDP supporting $\mathcal{D}$ shares the same state space, which may not hold in practice. We stress that we will prove a lower bound under Weak Proximity, showing that efficient generalization is impossible even in the simpler regime of a shared state space.

We now present Strong Proximity, a condition which strictly strengthens Weak Proximity. We will later show that unlike its Weak counterpart, Strong Proximity indeed permits efficient generalization (when the environments are deterministic).

**Condition 3 (Strong Proximity)** *The distribution $\mathcal{D}$ satisfies Strong Proximity with parameters $\xi_r, \xi_{tr}, \alpha \geq 0$ when:*

(a) *$\mathcal{D}$ satisfies Condition 1 with parameters $\xi_r, \xi_{tr} \geq 0$.*

(b) *There exists a deterministic policy $\pi^\star$ which is a near optimal policy for each MDP. Concretely, the policy $\pi^\star$ satisfies $V_M^s(\pi^\star) \geq \max_{\pi'} V_M^s(\pi') - \alpha$ for each state $s$ and each MDP $M$.*

Let us compare Weak with Strong Proximity. Part (a) remains identical. But Weak Proximity (b) only requires a shared policy which provides $\alpha$-suboptimal value with respect to $s_0$. This is in contrast to the shared policy in part (b) of Strong Proximity, which provides $\alpha$-suboptimal value for *any* state.

### 3.3 Tractability of Individual Optimization

As discussed previously, in order to efficiently solve Eqs. (1) & (2), we require the property that each individual MDP supporting $\mathcal{D}$ can be efficiently solved. It is natural to expect such a property to hold in practice. For instance, in the context of our earlier example of simulated robotic goal reaching tasks [48], any individual task can be efficiently solved via policy gradient methods. We now state two such properties, the first of which is strictly stronger than the second. Since these properties require a notion of query cost, we state both of them with reference to a generic query model QM, and when we later present our results we will instantiate QM to be either SQM or WQM. To avoid complicating notation in these statements, we assume in this subsection (as is our focus throughout

the paper) that all MDPs supporting $\mathcal{D}$ are defined on the same state-action space $\mathcal{S} \times \mathcal{A} \subset \mathbb{R}^d$. Recall that a linear policy $\pi$ is parameterized by $\overline{\theta} = \{\theta_h\}_{h=0}^{H-1}$, where $\theta_h \in \mathbb{R}^d$ and $\|\theta_h\|_2 = 1$ for all $0 \leq h \leq H-1$, such that $\pi(s) \in \operatorname{argmax}_{a \in \mathcal{A}}(s,a)^T \theta_h$ for any $s \in \mathcal{S}_h$. Here $x^T y$ denotes the Euclidean inner product of $x, y \in \mathbb{R}^d$. We use $\pi_M^\star$ to denote an arbitrary deterministic optimal policy of MDP $M$.

**Property 1 (Strong Individual Optimization (SIO))** *Let the query model be QM. The distribution $\mathcal{D}$ satisfies SIO with parameters $k > 0$ and $0 \leq \beta < 1/4$ when:*

*(a) Any MDP $M$ supporting $\mathcal{D}$ admits an optimal linear policy. Concretely, given any $M$, there exists $\overline{\theta^\star} = \{\theta_h^\star\}_{h=0}^{H-1}$ such that for every state $s \in \mathcal{S}_h$ we have $\pi_M^\star(s) \in \operatorname{argmax}_{a \in \mathcal{A}}(s,a)^T \theta_h^\star$.*

*(b) There exists a fixed and known algorithm, such that given any MDP $M$ and any state $s$, this algorithm uses at most $\mathcal{O}(|\mathcal{A}|H^k)$ query cost (under QM) on $M$ to identify (almost surely) a linear policy $\pi(\overline{\theta})$ parameterized by $\overline{\theta} = \{\theta_h\}_{h=0}^{H-1}$ which satisfies $\max_{\pi'} V_M^s(\pi') \geq V_M^s(\overline{\theta}) \geq \max_{\pi'} V_M^s(\pi') - \beta$. This algorithm then outputs $\pi(\overline{\theta})$ as well as $V_M^s(\overline{\theta})$.*

Let us discuss this property. Part (a) requires that for any MDP supporting $\mathcal{D}$, there exists an optimal linear policy. Part (b) requires that the user has knowledge of an algorithm, which can efficiently find a linear policy providing $\beta$-suboptimal value from any input state $s$ in any MDP $M$. The exponent $k$ describes the (polynomially sized) complexity of this algorithm. We stated the SIO property with respect to a generic efficient algorithm, since MDPs with different structures can require different types of algorithms to solve efficiently. Nevertheless, in our lower bound construction, the algorithm we provide to satisfy SIO is extremely simple and natural. It is simply a greedy version of Monte Carlo Tree Search, which is extremely popular in practice [29, 40].

SIO is a fairly strong property, since it says that a linear policy is sufficient to optimize any individual MDP, whereas in practice one typically requires nonlinear neural network policies. SIO also heavily constrains each individual MDP supporting $\mathcal{D}$ to be efficiently solvable from any initial state. We stress that we will prove our *lower bounds* under SIO, which makes our result stronger. Meanwhile, we prove our *upper bounds* under the following property, which is significantly weaker than SIO.

**Property 2 (Weak Individual Optimization (WIO))** *Let the query model be QM. The distribution $\mathcal{D}$ satisfies WIO with parameter $0 \leq \beta < 1/4$ when the following holds. There exists an oracle $\widehat{V}$, which takes as input a state $s$ and MDP $M$, and outputs $\widehat{V}_M^s$ satisfying $\max_{\pi'} V_M^s(\pi') \geq \widehat{V}_M^s \geq \max_{\pi'} V_M^s(\pi') - \beta$, via query cost (under QM) on $M$ that is polynomial in $|\mathcal{A}|, H, d$.*

WIO postulates the existence of an oracle $\widehat{V}$, which can efficiently approximate the optimal value that is achievable from an input state and MDP. To see that WIO is strictly weaker than SIO, simply note we can implement $\widehat{V}$ by running the algorithm described in part (b) of SIO. Note that in certain states, a user may use domain knowledge to implement $\widehat{V}$ without solving an entire RL problem. Also note that WIO does not place (arguably unrealistic) linearity restrictions on the MDPs supporting $\mathcal{D}$.

# 4 Main Results

We shall present our results in two subsections. In Section 4.1, we prove lower bounds which demonstrate that even under Weak Proximity, SQM and SIO, tractable generalization is worst case impossible. In Section 4.2, we prove that efficient generalization is possible under Strong Proximity, WQM and WIO, when the MDPs supporting $\mathcal{D}$ share a deterministic transition function.

## 4.1 Lower Bounds

Before stating our own results, we first state the following classical result which is known as the Simulation Lemma [26, 27, 6, 25, 1]. Recall that $\xi_r, \xi_{tr}$ are parameters used to satisfy Condition 1.

**Lemma 1** *Consider any $\mathcal{D}$ satisfying Condition 1 with $\xi_r, \xi_{tr} \geq 0$. For any policy $\pi$ and any $M_1, M_2$ supporting $\mathcal{D}$, we have that $|V_{M_1}^{s_0}(\pi) - V_{M_2}^{s_0}(\pi)| \leq \xi_r H + \xi_{tr} H$.*

This result is almost identical to the one given by [25], although there are some (minor) differences in assumptions so we provide a proof in Appendix D. This lemma shows that when $\mathcal{D}$ satisfies Condition 1 and $\xi_r, \xi_{tr}$ are each $o(\frac{1}{H})$, then efficient generalization is trivial, at least in problems where $H$ is large and we want to optimize to within $o(1)$ tolerance. Concretely, take any $M$ supporting $\mathcal{D}$ and use a standard RL method to find $\pi$ which satisfies $V_M^{s_0}(\pi) \approx \max_{\pi'} V_M^{s_0}(\pi')$. Then Lemma 1 ensures $V_{M'}^{s_0}(\pi) \gtrsim \max_{\pi'} V_{M'}^{s_0}(\pi') - o(1)$ for any other MDP $M'$ supporting $\mathcal{D}$. This implies $\mathbb{E}_{M \sim \mathcal{D}}[V_M^{s_0}(\pi)] \gtrsim \max_{\pi'} \mathbb{E}_{M \sim \mathcal{D}}[V_M^{s_0}(\pi')] - o(1)$ and $V_{M_{test}}^{s_0}(\pi) \gtrsim \max_{\pi'} V_{M_{test}}^{s_0}(\pi') - o(1)$.

Since Weak Proximity implies Condition 1, Lemma 1 and all the above statements remain true when $\mathcal{D}$ satisfies Weak Proximity. Naturally then, in our settings it is only interesting to consider problems when at least one of either $\xi_r$ or $\xi_{tr}$ is $\Omega(\frac{1}{H})$. Our next result is a *lower bound* which shows that when $\xi_r = \Theta(\frac{1}{H})$ and $\xi_{tr} = 0$, then Weak Proximity is not sufficient to efficiently generalize in the Average Performance Setting. For the statement of this result, recall that $\xi_r, \xi_{tr}, \alpha$ are parameters used to satisfy Weak Proximity, while $\beta, k$ are parameters used to satisfy SIO.

**Theorem 1** *Let the query model be SQM. For any $k \geq 3$, there exists $\mathcal{D}$ satisfying Weak Proximity with $\xi_r = \Theta(\frac{1}{H})$, $\xi_{tr} = 0$ & $\alpha = 0$ and SIO with $\beta = 0$ & $k$, such that the MDPs supporting $\mathcal{D}$ are deterministic and the following holds. Any (possibly randomized) algorithm requires $\Omega\left(\min\left\{|\mathcal{A}|^H, 2^d\right\}\right)$ total query cost to find (with probability at least $1/2$ over the randomness of the algorithm) a policy $\pi$ satisfying*

$$\mathbb{E}_{M \sim \mathcal{D}}[V_M^{s_0}(\pi)] \geq \max_{\text{linear policy } \pi'} \mathbb{E}_{M \sim \mathcal{D}}[V_M^{s_0}(\pi')] - 1/4.$$

We defer the proof to Appendix A.1. Let us discuss this theorem, which is stated for the Average Performance Setting, when the MDPs supporting $\mathcal{D}$ all share a deterministic transition function. Recall that SQM is the stronger query model we consider, which strengthens this lower bound, and trivially implies a lower bound for when WQM is the query model. Also recall that SIO is the stronger individual optimization property that we consider, and it ensures that the user can efficiently find a linear policy providing optimal value w.r.t any initial state for any individual MDP, since $\beta = 0$. Moreover, Weak Proximity (b) ensures that each MDP supporting $\mathcal{D}$ shares a policy that provides optimal value (w.r.t $s_0$), since $\alpha = 0$. And Weak Proximity (a) *explicitly* requires that the reward functions are (non-trivially) close, in the sense defined by Condition 1, because $\xi_r = \Theta(\frac{1}{H})$. Despite this significant structure, the theorem demonstrates that one can still require an exponential query cost to find a policy that is nearly as good as the best *linear* policy (which is of course easier than finding the best generic policy). Note that this lower bound holds with $\alpha = \beta = \xi_{tr} = 0$, and so implies a lower bound for when any of $\alpha, \beta, \xi_{tr}$ are strictly positive. As we discuss at the end of Section 4.1, Theorem 1 (and its forthcoming corollaries) immediately applies to the task of learning a feature mapping which maps similar states to the same vector, for the purpose of efficiently solving Average Performance and Meta RL settings.

We note that in the construction used to prove the lower bound of Theorem 1, the algorithm we provide to satisfy the SIO property is extremely simple and natural. It is simply a greedy version of Monte Carlo Tree Search, which is extremely popular in practice [29, 40].

Let us provide some intuition for our proof of Theorem 1. In the $|\mathcal{A}|$-ary tree hard instance used in our proof, there are $\Omega(|\mathcal{A}|^H)$ possible trajectories. The fact that $\xi_r = \Theta(\frac{1}{H})$ allows us enough degrees of freedom to hide the policy that generalizes across $\mathcal{D}$, so that identifying it requires querying each of the $\Omega(|\mathcal{A}|^H)$ trajectories. We leverage recent techniques [14, 45] to construct a suitable featurization of the state-action space, that is expressive enough to allow for efficiently finding an optimal linear policy for any single MDP, but does not leak any further information.

A similar result holds for the Meta RL setting. Recall that by SIO (b), the user has access to an algorithm which can solve any $M_{test}$ at test time in $\mathcal{O}(|\mathcal{A}|H^k)$ queries, *even if it does no training*. So it only makes sense to train, if one can use this training to solve $M_{test}$ in $o(|\mathcal{A}|H^k)$ queries. The following corollary to Theorem 1 demonstrates that this may require exponential query cost during training time. Its proof is presented in Appendix A.2.

**Corollary 1** *Let the query model be SQM. For any $k \geq 3$, there exists $\mathcal{D}$ satisfying Weak Proximity with $\xi_r = \Theta(\frac{1}{H})$, $\xi_{tr} = 0$ & $\alpha = 0$ and SIO with $\beta = 0$ & $k$, such that the MDPs supporting $\mathcal{D}$ are deterministic and the following holds. If a (possibly randomized) algorithm at test time can identify $\pi$ satisfying*

$$V_{M_{test}}^{s_0}(\pi) \geq \max_{\text{linear policy } \pi'} V_{M_{test}}^{s_0}(\pi') - 1/4$$

*in $o(|\mathcal{A}|H^k)$ queries, with probability at least $1/2$ over the selection of $\mathrm{M}_{\text{test}}$ (and the randomness of the algorithm), then this algorithm must have required $\Omega\left(\min\left\{|\mathcal{A}|^H, 2^d\right\}\right)$ total query cost at training time.*

So far we have presented results for when the MDPs supporting $\mathcal{D}$ share a deterministic transition function but have (slightly) varying rewards. For the remainder of Section 4.1, we present analogous results for when the MDPs share a reward function but have (slightly) varying transitions, again under both SIO and SQM. Recall from our discussion of Lemma 1 that when $\xi_{\mathrm{r}} = 0$, it is only interesting to consider problems when $\xi_{\mathrm{tr}}$ is $\Omega(\frac{1}{H})$. Unfortunately, our next result is a corollary of Theorem 1 which shows that efficiently solving the Average Performance Setting is impossible in this regime.

**Corollary 2** *Let the query model be SQM. For any $k \geq 3$, there exists $\mathcal{D}$ satisfying Weak Proximity with $\xi_{\mathrm{r}} = 0$, $\xi_{\mathrm{tr}} = \Theta(\frac{1}{H})$ & $\alpha = 0$ and SIO with $\beta = 0$ & $k$, such that the following holds. Any (possibly randomized) algorithm requires $\Omega\left(\min\left\{|\mathcal{A}|^H, 2^d\right\}\right)$ total query cost to find (with probability at least $1/2$ over the randomness of the algorithm) a policy $\pi$ satisfying*

$$\mathbb{E}_{M \sim \mathcal{D}}\left[V_M^{s_0}(\pi)\right] \geq \max_{\text{linear policy } \pi'} \mathbb{E}_{M \sim \mathcal{D}}\left[V_M^{s_0}(\pi')\right] - 1/4.$$

We defer the proof to Appendix A.3. We recall the discussion of Theorem 1, and note that the same discussion applies here, after swapping $\xi_{\mathrm{tr}}$ with $\xi_{\mathrm{r}}$. An analogous result holds for the Meta RL setting. As we discussed before presenting Corollary 1, it only makes sense to train, if one can use this training in order to solve $\mathrm{M}_{\text{test}}$ in $o(|\mathcal{A}|H^k)$ queries. The following result shows that this is impossible without exponential total query cost at training time. Its proof is presented in Appendix A.4.

**Corollary 3** *Let the query model be SQM. For any $k \geq 3$, there exists $\mathcal{D}$ satisfying Weak Proximity with $\xi_{\mathrm{r}} = 0$, $\xi_{\mathrm{tr}} = \Theta(\frac{1}{H})$ & $\alpha = 0$ and SIO with $\beta = 0$ & $k$, such that the following holds. If a (possibly randomized) algorithm at test time can identify $\pi$ satisfying*

$$V_{\mathrm{M}_{\text{test}}}^{s_0}(\pi) \geq \max_{\text{linear policy } \pi'} V_{\mathrm{M}_{\text{test}}}^{s_0}(\pi') - 1/4$$

*in $o(|\mathcal{A}|H^k)$ queries, with probability at least $1/2$ over the selection of $\mathrm{M}_{\text{test}}$ (and the randomness of the algorithm), then this algorithm must have required $\Omega\left(\min\left\{|\mathcal{A}|^H, 2^d\right\}\right)$ total query cost at training time.*

In conjunction with Lemma 1, the results of Theorem 1 and Corollaries 1, 2 & 3 suggest that the classical (and quite natural) way of measuring variation in MDPs using Condition 1 is not the right metric for the modern Average Performance & Meta RL settings. When both $\xi_{\mathrm{r}}$ and $\xi_{\mathrm{tr}}$ are $o(\frac{1}{H})$, then these settings are trivially solvable. But when either $\xi_{\mathrm{r}}$ or $\xi_{\mathrm{tr}}$ is $\Theta(\frac{1}{H})$ then these settings become exponentially hard, even under the additional Weak Proximity condition as well as SIO & SQM.

Note that Theorem 1 and Corollaries 1, 2 & 3 all hold in the setting where each MDP supporting $\mathcal{D}$ shares a state-action space. So these lower bounds immediately apply to more complex settings where the MDPs are defined on disjoint state-action spaces, and where learning an appropriate representation is necessary. Indeed, it is popular in practice to learn a feature mapping which maps similar states to the same vector. Our results show that if such a mapping enables efficient solution of the Average Performance & Meta RL settings, then learning the mapping itself is worst case inefficient.

## 4.2 Upper Bound

We now show that Strong Proximity permits efficient generalization when the MDPs supporting $\mathcal{D}$ share deterministic transitions. While this setting is restricted, we study it because our Theorem 1 shows that even this setting can be worst case inefficient under strong assumptions. Furthermore, past literature on even traditional RL with a single MDP has often focused on the deterministic setting [47, 15]. Notably, to prove our upper bound we only require the weaker WQM and weaker WIO. Our method is defined in Algorithm 1. It exploits Strong Proximity, which requires the existence of a policy which provides optimal value for each MDP from *any* given initial state, even though the objectives in Eqs. (1) & (2) are defined only in terms of value w.r.t $s_0$.

Let us describe Algorithm 1. It represents policy $\pi$ as a vector which stores one action for each timestep in $\{0, 1 \ldots H - 1\}$. It initializes arbitrary $\pi$ and incrementally updates it at each timestep

---
**Algorithm 1** Inputs: horizon length $H$, distribution $\mathcal{D}$, sample size $n$, oracle $\widehat{V}$ as defined in WIO
---
 1: Initialize $\pi$ as an arbitrary function from $\{0, 1 \ldots H - 1\}$ to $\mathcal{A}$
 2: **for** $t \in \{0, 1 \ldots H - 1\}$ **do**
 3:     **for** $i \in \{1, 2 \ldots n\}$ **do**
 4:         Sample $M_i \sim \mathcal{D}$
 5:         **for** $a \in \mathcal{A}$ **do**
 6:             Begin a new episode in $M_i$ at $s_0$
 7:             **if** $t > 0$ **then** Execute action sequence $\{\pi(t')\}_{0 \leq t' < t}$ to arrive at $s_t$ **end if**
 8:             Take action $a$ to arrive at $s' = \mathcal{T}_{M_i}(s_t, a)$ and receive $R_{M_i}(s_t, a)$
 9:             Query $\widehat{V}$ to obtain $\widehat{V}_{M_i}^{s'}$ and store $Q_{i,a} = R_{M_i}(s_t, a) + \widehat{V}_{M_i}^{s'}$
10:         **end for**
11:     **end for**
12:     Store $a_t \in \operatorname{argmax}_{a' \in \mathcal{A}} \{\frac{1}{n} \sum_{i=1}^{n} Q_{i,a'}\}$ and define $\pi(t) = a_t$
13: **end for**
14: **return** $\pi$
---

$t$. At the beginning of any timestep $t > 0$, $\pi$ has been constructed to play the action $\pi(t') = a_{t'}$ at each timestep $t' < t$. The algorithm then executes $\{\pi(t')\}_{0 \leq t' < t}$ to arrive at $s_t$. Crucially, due to the assumption of a shared state-action space and shared deterministic transitions, the state $s_t$ is fully determined by $\pi$ and does *not* depend on the particular $M_i$. Exploiting WIO, the method queries $\widehat{V}$ to estimate the value in $M_i$ of each child state of $s_t$. Averaging this estimated value over $\{M_i\}_{i=1}^{n}$ yields an estimate of the expected value (over the randomness in $\mathcal{D}$) of each action at $s_t$. Finally, the algorithm picks the action $a_t$ with the highest estimated value, and updates $\pi$ to play $a_t$ at timestep $t$. This algorithm operates in the standard RL framework and falls under the purview of WQM.

The following result provides a performance guarantee for Algorithm 1. Recall that $\alpha, \xi_{\mathrm{tr}}, \xi_{\mathrm{r}}$ are parameters used to satisfy Strong Proximity and $\beta$ is a parameter used to satisfy WIO.

**Theorem 2** *Let the query model be WQM. Consider any $\mathcal{D}$ satisfying WIO with $\beta \geq 0$ and Strong Proximity with $\xi_{\mathrm{tr}} = 0$ and any $\alpha, \xi_{\mathrm{r}} \geq 0$, such that the MDPs supporting $\mathcal{D}$ are deterministic. Fix $\epsilon, \delta > 0$, and let $\pi$ be the output of Algorithm 1 when run with $n = \frac{H^2}{\epsilon^2} \log\left(\frac{2H|\mathcal{A}|}{\delta}\right)$ samples. Then with probability at least $1 - \delta$, we are guaranteed that*

$$\mathbb{E}_{M \sim \mathcal{D}}[V_M^{s_0}(\pi)] \geq \max_{\pi'} \mathbb{E}_{M \sim \mathcal{D}}[V_M^{s_0}(\pi')] - \epsilon - 3\alpha H - 3\beta H.$$

*Hence the total query cost under WQM required to achieve this guarantee is polynomial in $q_{\mathcal{D}}, |\mathcal{A}|, H, d$.*

We defer the proof to Appendix B. A few comments are in order. First, note that Theorem 2 directly provides a guarantee for the Average Performance setting. It also provides a guarantee for the Meta RL setting, since the $\pi$ found by Algorithm 1 will on average perform well for $\mathrm{M}_{\mathrm{test}}$, and the user can use $\pi$ to warm start any finetuning or adaptation at test time. Second, the specified value of $n$ depends only on quantities that are either known a priori or chosen by the user. This makes Algorithm 1 *parameter free* — the user does not need to know the values of $\alpha, \beta, \xi_{\mathrm{r}}, \xi_{\mathrm{tr}}$ to run this method.

Third, note Theorem 2 holds under WIO. By contrast, Weak Proximity was insufficient for efficient generalization even when paired with SIO. This suggests that a condition that is both necessary and sufficient for efficient generalization lies somewhere between Weak and Strong Proximity — assuming, of course, that we do not assume an individual optimization property that is even stronger than SIO. Indeed, SIO is already quite strong, since SIO says that a linear policy is sufficient to optimize any individual MDP, but in practice one typically employs nonlinear neural network policies.

Finally, observe that $\xi_{\mathrm{r}}$ does not appear in the error bound. So $\xi_{\mathrm{r}}$ can be arbitrarily large, and Theorem 2 requires no *explicit* conditions on the reward functions of the MDPs supporting $\mathcal{D}$, as in the sense of Condition 1. Instead, the *implicit* reward structure induced by the shared nearly optimal policy required by Strong Proximity is sufficient. Comparing this observation with the result of Theorem 1 suggests that the classical explicit constraints on rewards and transitions is not appropriate for modern RL generalization settings. Instead, implicit constraints of the sort afforded by Strong Proximity offer a more fine grained characterization of when efficient generalization is possible.

Recall from Theorem 1 that a lower bound holds for Weak Proximity and SIO even with $\alpha = \beta = 0$. However, Strong Proximity and WIO provide enough structure that the error bound of Theorem 2 can tolerate $\alpha, \beta \geq 0$. But these $\alpha, \beta$ terms in the error bound of Theorem 2 scale linearly with $H$. It is natural to question whether this scaling is due to a suboptimality of Algorithm 1 or looseness in our analysis. We provide a partial answer to this question in Appendix C, where we prove that the dependency on $\beta$ given in the result of Theorem 2 is tight to within a logarithmic factor in $H$.

## 5 Discussion

In this paper, we studied the design of RL agents that generalize. We proved that efficient generalization is worst case impossible, even under structural conditions like Weak Proximity and strong assumptions on the query model and tractability of individual MDPs. This result extends to the task of learning representations for the purpose of efficient generalization. On the positive side, we provided Strong Proximity, which permits efficient generalization, even under mild assumptions on the query model and individual tractability. Our analysis highlights that classical metrics for measuring similarity of MDPs are inappropriate for modern RL. It also suggests that a condition which is both necessary and sufficient for efficient generalization lies between Weak & Strong Proximity — unless we make (arguably unreasonable) assumptions on the tractability of individual MDPs.

**Negative Societal Impacts.** Our work is theoretical, and we do not foresee any direct societal impacts, at least in the short term. In the long term, our work may increase the technological feasibility of developing agents that can be deployed in society. In this scenario, a bad actor may deploy harmful, malicious agents. This must be prevented by properly understanding the technology (which our work aims to do), and working with policy makers to prevent bad actors from accessing it.

**Limitations of Our Work.** The primary limitation of our work is that our upper bound has limited applicability. It holds only when the MDPs share a state-action space, and when the MDPs are determinstic, which is very restrictive in practice. Our rationale for working in this restricted setting was due to our lower bounds, which show that even this toy setting can be worst case inefficient, and because it is necessary to understand the toy setting before looking at more complex scenarios. Nevertheless, our upper bound is several steps removed from the practice of RL. It is best interpreted as a preliminary sufficient condition for when efficient generalization is possible, albeit in a toy setting, and is far from conclusive on this matter.

**Future Work.** Note that our upper bound might apply if we are a priori given a feature mapping which maps similar states of different MDPs to the same state space. For example, in self driving, learning to drive in different countries might be difficult because the images of traffic signs are different. But if a known feature map extracts the underlying meaning of these signs, then our upper bound could conceivably apply. The key direction for future work, is how to learn such a feature mapping efficiently, while ensuring that it is still useful for generalization.

## Acknowledgments and Disclosure of Funding

This material is based upon work supported by the National Science Foundation Graduate Research Fellowship Program under Grant No. DGE1745016. We further acknowledge the support of NSF via RI 2007517 and IIS-1909816. Any opinions, findings, and conclusions or recommendations expressed in this material are those of the authors and do not necessarily reflect the views of the National Science Foundation.

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
