Figure 1: An illustration of the generic binary tree structure used to define the MDPs that support the $\mathcal{D}$ constructed in the proof of Theorem 1.

## A Lower Bound Proof Details

In this section, we will provide proofs of Theorem 1 and Corollaries 1, 2 & 3. For ease in presentation, we shall assume throughout that the action space $\mathcal{A}$ for each MDP contains two actions, which we denote $a_1$ and $a_2$. It is easy to extend the proofs to the case when there are many actions. We will often use the notation $s_1$ and $s_2$ to denote the child states of a state $s$ when taking actions $a_1$ and $a_2$ respectively. We shall also use $\pi_M^*$ to denote an optimal policy for MDP $M$. Whenever we require $H$ to be sufficiently large for an algebraic argument to go through, we shall assume so.

### A.1 Proof Of Theorem 1

**Construction Details.** Our construction of $\mathcal{D}$ will consist of MDPs whose shared state and action spaces are generically defined by a binary tree. The structure of the binary tree is depicted in Figure 1. The tree is of length $H$, each node in the tree will define a different state and the edges connecting two nodes define the actions. In this fashion, each state has two actions, which we denote $a_1$ and $a_2$, and taking either action leads deterministically to the corresponding child. Taking any action from states on the final level of the tree exits the MDP. The state $s_0$ is the root of the tree. We thus have defined the shared state and action space for MDPs supporting $\mathcal{D}$, and have also defined the shared and deterministic transition dynamics. So the MDPs are all deterministic by construction. Note that to verify Weak Proximity (a), the construction so far implies that the state-action space is shared and the transitions satisfy $\xi_{\mathrm{tr}} = 0$, although we have not yet defined the reward structure so cannot say anything about $\xi_{\mathrm{r}}$.

To complete the definition of $\mathcal{D}$, we must complete the definition of each individual MDP supporting $\mathcal{D}$ by defining a reward function for each MDP. We shall do so via the procedure described below. But before that, fix a state $s^*$ on level $H$, which is selected by sampling uniformly at random from the set of states on level $H$. Note that the location of $s^*$ will be kept hidden from the user, although we will define $\mathcal{D}$ with reference to a fixed $s^*$.

Note that there are $2^{\frac{H}{2}-1} - 1$ states $s$ on level $\frac{H}{2}$ such that the subtree rooted at $s$ does not contain $s^*$. Also note that each subtree rooted at such an $s$ has $2^{\frac{H}{2}-1}$ states on level $H$. If one were to "view" the final level of a binary tree on paper, as in Figure 1, there is a natural ordering of the states on the final level from left to right. In each subtree rooted at such an $s$, consider the $i$th state in this ordering of the states on the final level of that subtree, and denote this $i$th state as $x_{i,s}$. Then define $\mathcal{S}_i = \cup_s x_{i,s}$. There are a total of $2^{\frac{H}{2}-1}$ such sets $\mathcal{S}_i$. We will construct $2^{\frac{H}{2}-1}$ MDPs total to support the distribution $\mathcal{D}$, one for each $\mathcal{S}_i$. Note that for $i \neq j$, we have $\mathcal{S}_i \cap \mathcal{S}_j = \varnothing$. Note also that $\mathcal{S}_i$ never contains a state which is also in the subtree rooted at level $\frac{H}{2}$ that contains $s^*$.

Define $\epsilon = \frac{1}{\frac{H}{2}-\log(H^k)-1}$, and note that $\epsilon$ is $\Theta(\frac{1}{H})$. For each $\mathcal{S}_i$, we will define the MDP $M_i$ by defining $R_{M_i}$ as follows. Fix an $\mathcal{S}_i$. For each $s \in \mathcal{S}_i$, let $s_p$ denote the unique ancestor of $s$ on level

$\frac{H}{2} + \log(H^k) + 1$ of the tree. On the path connecting $s_p$ to $s$, assign each state-action pair a reward of $\epsilon$. So the total reward accumulated by following the path from the root through $s_p$ to $s$ and taking either action from $s$ is $\epsilon(\frac{H}{2} - \log(H^k) - 1) = 1$. Also, consider the unique ancestor $s_p^*$ of $s^*$ on level $\frac{H}{2} + \log(H^k) + 1$, and assign each state-action pair along the path $s_p^*$ to $s^*$ a reward of $\epsilon$. So the total reward accumulated by following the path from the root to $s^*$ and taking either action from $s^*$ is $\epsilon(\frac{H}{2} - \log(H^k) - 1) = 1$. Any other state-action pair in the tree is assigned zero reward. This completes the definition of $R_{M_i}$, and hence the definition of $M_i$, except that we have not yet featurized the state-action space.

Perform this procedure for each of the $2^{\frac{H}{2}-1}$ sets $\mathcal{S}_i$, to obtain a set which contains $2^{\frac{H}{2}-1}$ such MDPs. Once we featurize the state-action space in the fashion described below, the definition of $\mathcal{D}$ is completed by assigning the uniform distribution to this set. The key behind this construction, is that for any MDP supporting $\mathcal{D}$, each subtree rooted at level $\frac{H}{2}$ contains a single path which provides the optimal unit value. But there is only one path providing unit value that is shared by each MDP, this is the path from the root to $s^*$. Note that for any $M_i$, all of the subtrees rooted at states on level $\frac{H}{2}$ are identical, with the exception of the subtree containing $s^*$.

Let us now discuss how to featurize the state-action space. Note that each MDP is defined on a common state-action space that has at most $2^H$ states. Sample vectors $\{z_1, z_2 \ldots z_{2^H}\}$ i.i.d from the spherical measure on the surface of the unit sphere that sits in $d - 2$ dimensions. By the results of [14], we can pick $d$ to be at most a polynomial of $H$, while ensuring that $\|z_i\|_2 = 1$ and $|z_i^T z_j| \leq \frac{1}{50}$ for all $i \neq j$. Take any ordering of the states $\{s_i\}$ indexed by $i$, and assign the following features

$$\phi(s_i, a_1) = [z_i, 1, 0] \text{ and } \phi(s_i, a_2) = [0, 1, 1].$$

Rescaling the features to have unit norm is trivial, so we work with these since they make the computations more apparent. In the above, it appears that $\phi(s_i, a_2)$ is the same for each $i$, but this is without loss of generality since we can always add a dummy coordinate that makes them all unique. Crucially note that the features do *not* depend on the reward structure of the MDP, since they are completely agnostic to the choice of rewards. Hence they do not leak any information about the rewards.

**Verifying Weak Proximity (a).** We have already checked above that the state-action space is shared and $\xi_{tr} = 0$. To see that $\xi_r$ is $\Theta(\frac{1}{H})$, simply note that any state in any MDP supporting $\mathcal{D}$ has reward either $0$ or $\epsilon$, and $\epsilon$ is $\Theta(\frac{1}{H})$. This verifies Weak Proximity (a).

**Verifying Weak Proximity (b).** Define $\pi^*$ to be the deterministic policy which prescribes the path leading from the root to $s^*$, and at states not along this path it prescribes an arbitrary action. It is then immediate from our above arguments that for any $M$, $V_M^{s_0}(\pi^*) = \max_{\pi'} V_M^{s_0}(\pi') = 1$, so Weak Proximity (b) is satisfied with parameter $\alpha = 0$.

**Verifying SIO (a).** Fix any $M$. The key to verifying this is the observation that all subtrees rooted at level $H/2$ are identical (in terms of reward structure, not features), except for the one that contains $s^*$. In the first $H/2 - 1$ levels one can use an arbitrary policy. So now consider any level $h$ that is greater than or equal to $H/2$. Inside the subtree rooted at $H/2$ that contains $s^*$, there is a unique state $s$ on this level that is an ancestor of $s^*$. Let us denote $z$ to be the spherical measure random variable that was used to define the feature map for this state, so that $\phi(s, a_1) = [z, 1, 0]$ and $\phi(s, a_2) = [0, 1, 1]$.

Now observe that within the subtree rooted at $H/2$ that contains $s^*$, the only state from where one can achieve nonzero reward (on the level $h$ we are considering) is $s$. So the policy at the other states in this subtree does not matter. Similarly, in each of the other subtrees rooted at $H/2$, there is a unique state $s'$ which can yield nonzero reward (on the level $h$ we are considering). Furthermore, since these other subtrees are identical, the optimal policy within one subtree works for another. To construct our $\theta \equiv \theta_h^*$ for this level $h$, we use this information to claim that there are four cases to consider.

First, consider the case when $\pi_M^*(s) = a_1$ and $\pi_M^*(s') = a_2$ for any $s'$ in another subtree which can yield nonzero reward. Let $\theta = [z, 0, 1/2]$. Then we have that

$$\phi(s, a_1)^T \theta = [z, 1, 0]^T [z, 0, 1/2] = 1 + 0 + 0 = 1 \text{ and } \phi(s, a_2)^T \theta = [0, 1, 1]^T [z, 0, 1/2] = 0 + 0 + 1/2 = 1/2,$$

which implies that we pick $a_1$ at $s$ if we follow the linear policy. But

$$\phi(s', a_1)^T \theta = [z', 1, 0]^T [z, 0, 1/2] \leq \frac{1}{50} + 0 + 0 = \frac{1}{50} \text{ and } \phi(s', a_2)^T \theta = [0, 1, 1]^T [z, 0, 1/2] = 0 + 0 + 1/2 = 1/2,$$

which implies that we pick $a_2$ at $s'$ if we follow the linear policy $\theta$. Note that this argument holds for *any* $s'$ which can yield nonzero reward lying in *any* of the subtrees that do not contain $s^*$, and also recall the optimal policy does not change when looking at different subtrees not containing $s^*$.

Second, consider the case when $\pi_M^*(s) = a_2$ and $\pi_M^*(s') = a_1$ for any $s'$ in another subtree which can yield nonzero reward. Let $\theta = -[z, 0, 1/2]$. Since we have simply negated the above case, the result holds.

Third, consider the case when $\pi_M^*(s) = a_1$ and $\pi_M^*(s') = a_1$ for any $s'$ in another subtree which can yield nonzero reward. Let $\theta = [0, 1, -1/2]$. Then the result follows immediately because $\phi(s'', a_1)^T \theta = 1$ but $\phi(s'', a_2)^T \theta = 1/2$ for every state $s''$ in the tree.

Fouth, consider the case when $\pi_M^*(s) = a_2$ and $\pi_M^*(s') = a_2$ for any $s'$ in another subtree which can yield nonzero reward. Let $\theta = -[0, 1, -1/2]$. Then the result follows immediately because we have simply negated the above case.

It remains to rescale the features and $\theta$ to ensure they all have unit norm.

**Verifying SIO (b).** Consider the following algorithm, which takes as input an MDP $M_i$ that is known to support $\mathcal{D}$, but it knows nothing else about $M_i$ (in particular it does not know which $\mathcal{S}_i$ was used to define $M_i$). It of course does not know anything about the location of $s^*$ other than the fact that $s^*$ was sampled uniformly at random to define $\mathcal{D}$. It has also not been allowed to query any other MDPs supporting $\mathcal{D}$. The algorithm begins by picking an arbitrary state $s'$ on level $\frac{H}{2}$. It then queries each state on level $\log(H^k) + 1$ of the subtree rooted at $s'$ (this is level $\frac{H}{2} + \log(H^k) + 1$ of the entire tree). There is a single state $s_p$ on this level which has an action providing reward $\epsilon$, and all other states will give reward zero. When it finds the state $s_p$ on this level providing reward $\epsilon$, it stores the path leading to this state $s_p$. It then queries each of the two child states of $s_p$ to identify which of these child states lies along the optimal path, which is doable since exactly one of these child states has an action which provides reward $\epsilon$. It greedily takes the action required to arrive at this child, and stores this action. It then repeats this greedy procedure from the child $\Theta(H)$ times until it reaches the final level of the tree. In this manner, if it was given MDP $M_i$ as input then it identifies a path from the root through $s'$ and $s_p$ (or perhaps $s_p^*$) to either $s^*$ or some state $s$ on level $H$ which satisfies $s \in \mathcal{S}_i$. Let $\pi$ be the deterministic policy which prescribes this path, and prescribes arbitrary actions for states not along this path, then it is clear $V_{M_i}^{s_0}(\pi) = \max_{\pi'} V_{M_i}^{s_0}(\pi') = 1$. Note also that this same algorithm also immediately can be used to find $\pi$ satisfying $V_{M_i}^a(\pi) = \max_{\pi'} V_{M_i}^a(\pi')$ for any initial state $a$. To see this, note that if $a$ is an ancestor of $s^*$ or some $s \in \mathcal{S}_i$, then the described algorithm will identify this and find an appropriate path. If $a$ is not an ancestor of either of these, then the value of this state is zero and the same algorithm can be used to certify this. The sample complexity of this algorithm is the number of queries it took at the beginning to identify $s_p$ in the subtree rooted at $s'$. Note that there are $2^{\log(H^k)+1}$ states on level $\log(H^k) + 1$ of the subtree rooted at $s'$, so the sample complexity of this algorithm is $\mathcal{O}(H^k)$. To convert the policy $\pi$ found by the algorithm into a linear policy $\bar{\theta} = \{\theta_h\}_{h=0}^{H-1}$, simply note that we can set $\theta_h = [0, 1, -\frac{1}{2}]$ if $\pi$ recommends $a_1$ while following its path at timestep $h$, and set $\theta_h = [0, -1, \frac{1}{2}]$ if $\pi$ recommends $a_2$ while following its path at timestep $h$. Then rescale $\theta_h$ to ensure it has unit norm. This verifies SIO (b) with parameter $\beta = 0$.

With this construction of $\mathcal{D}$ in hand, we return to the proof of Theorem 1, during which we shall also prove the claim we made above about how $s^*$ cannot be identified in a polynomial number

of samples. A basic computation reveals that any path through the tree which does not end in $s^*$, has value (in expectation over $\mathcal{D}$ w.r.t. $s_0$) at most $\frac{2}{H}$. However, the path through the tree which ends in $s^*$ has value (in expectation over $\mathcal{D}$ w.r.t. $s_0$) precisely one. The optimal policy (in terms of value in expectation over $\mathcal{D}$ w.r.t. $s_0$) is hence clearly the deterministic policy which prescribes following the path through the tree that ends in $s^*$. Hence, any algorithm which can find a policy $\pi$ satisfying $\mathbb{E}_{M \sim \mathcal{D}}[V_M^{s_0}(\pi)] \geq \max_{\pi'} \mathbb{E}_{M \sim \mathcal{D}}[V_M^{s_0}(\pi')] - \frac{1}{4}$ must be able to identify $s^*$ with non-trivial probability. So to show that finding such a $\pi$ requires $\Omega(2^H)$ queries, it is sufficient to show that identifying $s^*$ requires $\Omega(2^H)$ queries.

This is immediate from the nature of our construction. Simply observe that any algorithm must query in the subtree rooted at level $\frac{H}{2}$ that contains $s^*$ at least once in order to identify $s^*$. However, for any MDP there are $2^{\frac{H}{2}-1}$ subtrees on level $\frac{H}{2}$ and all but one of them are identical. Of course, identifying the correct subtree with non-trivial probability requires $\Omega(2^H)$ total queries. Note that we crucially used here the fact that the features do *not* depend on the reward structure of the MDP, and hence do not leak any information about the rewards. Note also that our argument holds when the agent has access to a generative model, since it can transition to any state to query it. Moreover, the difficulty here comes from querying states, and not from sampling MDPs. So the result holds under SQM. Finally, we note that the policy prescribing the path through $s^*$ can easily be expressed as a linear policy. This completes the proof.

### A.2 Proof Of Corollary 1

We use the same $\mathcal{D}$ that was constructed in the proof of Theorem 1. Before proceeding with the proof of Corollary 1, we note the following key fact. If we sample $M$ from $\mathcal{D}$, but do not query any MDP supporting $\mathcal{D}$ beforehand, then any algorithm that can find (possibly nonlinear) $\pi$ satisfying $V_M^{s_0}(\pi) \geq \max_{\pi'} V_M^{s_0}(\pi') - 1/4$, with probability at least $1/2$ over the selection of $M$, requires $\Omega(H^k)$ query cost (under QM) on $M$. This fact demonstrates that the algorithm described in SIO (b) is minimax optimal, since *any* procedure will need the same complexity to solve an MDP sampled from $\mathcal{D}$, assuming it has not already queried other MDPs beforehand. In particular, any algorithm which can solve $\mathrm{M_{test}}$ at test time in $o(H^k)$ queries must rely on querying during training time.

To prove this fact, note that any algorithm (that is optimal to within the constant $\frac{1}{4}$) must discover a path that has $\Omega(H)$ state-action pairs intersecting with a path that leads either to $s^*$ or to a state $s \in \mathcal{S}_i$. Of course, discovering such a path is equivalent to identifying $s^*$ or the $s_p$ corresponding to $s \in \mathcal{S}_i$. Assume for now the claim that we cannot identify $s^*$ with a polynomial number of samples. Then we need only show that identifying an $s_p$ corresponding to $s \in \mathcal{S}_i$ requires $\Omega(H^k)$ queries. But this is immediate, since all we know is that $M_i$ was sampled from $\mathcal{D}$, and each subtree (that doesn't contain $s^*$) rooted at level $\frac{H}{2}$ is identical. So without loss of generality take any subtree (not containing $s^*$) rooted at level $\frac{H}{2}$, then a priori our prior for the location of $s_p$ in that subtree is exactly the uniform distribution over states on level $\log(H^k) + 1$ of that subtree. So we must query at least half of the $2^{\log(H^k)+1} = \Omega(H^k)$ states on level $\log(H^k) + 1$ of that subtree to identify $s_p$ with non-trivial probability. Conditioned on our claim that it is not easy to identify $s^*$, the claimed fact. Note that this argument uses the fact that we can directly query states on level $\log(H^k) + 1$ of that subtree, and so holds under a generative model. Note that we crucially used here the fact that the features do *not* depend on the reward structure of the MDP, and hence do not leak any information about the rewards.

We now return to the proof of Corollary 1. For any $M_i$, there are two types of paths that are optimal, the first is through $s^*$ and the others are through the states lying in $\mathcal{S}_i$. Note that since $\mathrm{M_{test}} \sim \mathcal{D}$ at test time, the location of the optimal paths in $\mathrm{M_{test}}$ that do not intersect $s^*$ are sampled uniformly at random. A nearly identical argument to the one used in the proof of Theorem 1, and the proof of the preceding fact, shows that identifying any of these optimal paths that are sampled uniformly at random requires $\Omega(H^k)$ queries. The only difference is that we condition on the event that $\mathrm{M_{test}}$ is not the same as any of the MDPs queried during training, which occurs with high probability when we are only allowed a polynomial number of queries during training time. Note that even after conditioning on this event, finding an optimal path (which does not go through $s^*$) of $\mathrm{M_{test}}$ requires

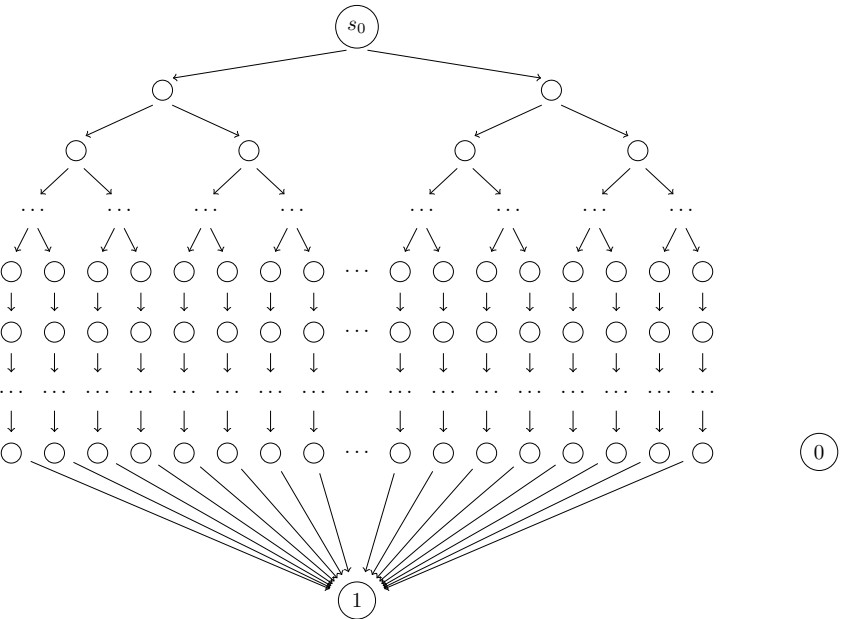

Figure 2: An illustration of the generic structure used to define the MDPs that support the $\mathcal{D}$ constructed in the proof of Corollary 2. Observe that the first half of the structure is a tree, while the second half comprises of linear sequences of states.

$\Omega(H^k)$ queries. This is simply because conditioning on the aforementioned event only changes the probability of the location of such a path by a value which is exponentially small in $H$, assuming again that we used polynomial in $H$ number of queries during training. So we cannot perform any inference to reduce the size of the set of feasible locations of an optimal path (which does not go through $s^*$). Then we can essentially repeat the same argument used in the proof of Theorem 1. Hence, if an algorithm hopes to solve $M_{\text{test}}$ at test time in $o(H^k)$ queries, then during training time it must narrow the possible locations of $s^*$ to a set whose cardinality is polynomial in $H$. But by the proof of Theorem 1, this would require $\Omega(2^H)$ queries.

### A.3 Proof Of Corollary 2

**Construction Details.** The $\mathcal{D}$ that we construct here is very similar in spirit to the $\mathcal{D}$ that was constructed in the proof of Theorem 1. $\mathcal{D}$ will be supported by MDPs whose shared state and action spaces all share the same generic structure. We depict this generic structure in Figure 2. For the first $\frac{H}{2}$ levels, the structure is defined by a binary tree, where the nodes in the tree represent states. Each state in the tree has two actions. For the next $\frac{H}{2}$ levels, there are numerous linear sequences of states of length $\frac{H}{2}$. Each such sequence emanates from a corresponding state that is a leaf of the tree. These sequences all end in a common state, which we denote $\textcircled{1}$. Each state in the sequence has a single action. The state $s_0$ is the root of the tree. There is also a state $\textcircled{0}$, which a priori is disconnected from the remainder of the structure. This implies that there is a shared state-action space. For each MDP $M$ supporting $\mathcal{D}$, we will have that $R_M$ maps $\textcircled{1}$ to one, and maps all other states including $\textcircled{0}$ to zero. Taking any action from $\textcircled{0}$ or $\textcircled{1}$ always exits the MDP for any MDP supporting $\mathcal{D}$. A priori, the transition function in this generic structure is deterministic. And it is "natural" in the sense that taking either action from a state in the tree leads to the corresponding child, and taking the action when in the linear sequence of states leads to the next state in the sequence. Of course, we will modify this "a priori natural" transition function in different ways for each MDP.

Let $\mathcal{F}_h$ denote the set of states at level $h$ of this generic structure that we have outlined above. Before constructing $\mathcal{D}$, first select a state $s^*$ uniformly at random from the $2^{H/2-1}$ states in $\mathcal{F}_{H/2}$. As in the

proof of Theorem 1, $s^*$ is hidden from the user.

Note that there are $2^{\frac{H}{4}-1} - 1$ states $s$ on level $\frac{H}{4}$ such that the subtree rooted at $s$ does not contain $s^*$. Also note that each subtree rooted at such an $s$ has $2^{\frac{H}{4}-1}$ states on level $\frac{H}{2}$ of the entire structure. If one were to "view" the final level of a binary tree on paper, there is a natural ordering of the states on the final level from left to right. In each subtree rooted at such an $s$ such that subtree does not contain $s^*$, consider the $i$th state in this ordering of the states on the final level of that subtree (which is the level $\frac{H}{2}$ of the entire structure). Denote this $i$th state as $x_{i,s}$. Then define $\mathcal{S}_i = \cup_s x_{i,s}$. There are a total of $2^{\frac{H}{4}-1}$ such sets $\mathcal{S}_i$. We will construct $2^{\frac{H}{4}-1}$ MDPs total to support the distribution $\mathcal{D}$, one for each $\mathcal{S}_i$. Note that for $i \neq j$, we have $\mathcal{S}_i \cap \mathcal{S}_j = \varnothing$. Note also that $\mathcal{S}_i$ never contains a state which is also in the subtree rooted at level $\frac{H}{4}$ that contains $s^*$.

To define each MDP $M$ that supports $\mathcal{D}$, it is sufficient to define the transition function $\mathcal{T}_M$ for $M$, since we have already defined the shared state-action space and the shared reward function. Similar to the proof of Theorem 1, for each $\mathcal{S}_i$ construct an MDP $M_i$ as follows:

1. Consider the linear portion of $M_i$ that lies below any state $s'$ satisfying $s' \notin \mathcal{S}_i$ and $s' \neq s^*$. Recall that a priori, the generic structure for any MDP was deterministic, which implies when you are at a state in this linear portion then taking the action deterministically leads to the next state in the sequence. We now modify this so that for any state-action pair along the linear sequence below $s'$, taking the action takes you to the child with probability $1 - 10/H$, and makes you jump out to state ⓪ with probability $10/H$. Do this for all $s'$ satisfying $s' \notin \mathcal{S}_i$ and $s' \neq s^*$. Leave the linear sequence of states below $s^*$ and any $s \in \mathcal{S}_i$ unchanged, so that the transitions in these two linear sequences remain deterministic.

2. For each $s \in \mathcal{S}_i$, there is a unique path connecting state $s$ to its unique ancestor on level $\frac{H}{4}$. This path defines a sequence of state-action pairs that leads one from level $\frac{H}{4}$ to $s$. Again recall that a priori, the generic structure for any MDP was deterministic, which implies that taking one of these actions while in the tree deterministically leads you to the corresponding child. Modify the transitions for these state-action pairs along this path of length $\frac{H}{4}$, so that when you take an action, then with probability $1 - 1/H^{k-1}$ the action leads to the child, and with probability $1/H^{k-1}$ it takes you directly to state ①. Do this for each $s \in \mathcal{S}_i$. Similarly, consider the path between $s^*$ and its unique ancestor on level $\frac{H}{4}$. Modify the transitions for the state-action pairs along this path so that when you take an action, then with probability $1 - 1/H^{k-1}$ the action leads to the child, and with probability $1/H^{k-1}$ it takes you directly to state ①.

We have thus defined $\mathcal{T}_{M_i}$ for each $M_i$, where each $M_i$ is identified by a particular $\mathcal{S}_i$. This defines a set containing a total of $2^{\frac{H}{4}-1}$ different MDPs $M_i$. To complete the definition of $\mathcal{D}$, simply consider the uniform distribution over the set of $M_i$ that we have created. We use a featurization that is identical to the one used in Theorem 1 for the binary tree portion of our construction here, and arbitrary features for the linear portion since there is only one action to take here.

**Verifying Weak Proximity (a).** We have already established the state-action space for each MDP is shared. Note that by definition each MDP shares a reward function, since ① always provides unit reward and each other state provides zero reward, regardless of the MDP. So we have $\xi_r = 0$. Then observe that when we modified the transitions at states, we changed them by making them lead to states ⓪ or ① with probability at most $\max\{\frac{1}{H^k}, \frac{10}{H}\}$. It remains to note the $\ell_1$ characterization of the total variation distance and that $k \geq 3$, implying that $\xi_{\mathrm{tr}} = \Theta(\frac{1}{H})$. This verifies Weak Proximity (a).

**Verifying Weak Proximity (b).** Observe that following the unique path leading from the root through state $s^*$ till ① always provides value 1, regardless of the MDP. This is because if we take the actions corresponding to this path, then either we hop out to ① while we are in the tree, or we

reach the linear sequence from where we deterministically arrive at $(1)$. Hence this path is always optimal. In turn, the policy which prescribes this path, while doing something arbitrary at states not along this path, always provides optimal value. So Weak Proximity (b) is satisfied with parameter $\alpha = 0$.

**Verifying SIO (a).** The proof of this is identical to the one in Theorem 1.

**Verifying SIO (b).** To verify this, sample $M_i \sim \mathcal{D}$ and consider the following greedy algorithm. Note that $s^*$ is not revealed a priori, nor do we know the locations of the states in $\mathcal{S}_i$, and the only information that the user has is that this MDP supports $\mathcal{D}$. Otherwise this problem trivially does not require any samples to solve. Start at any arbitrary state on level $\frac{H}{4}$ and sample the left action $\mathcal{O}(H^{k-1})$ times and the right action $\mathcal{O}(H^{k-1})$ times. With high probability, one of these actions will lead to the state $(1)$ at least once. And of course with unit probability the other action will not lead to $(1)$. Simply take the action that has led to $(1)$ at least once, and repeat this procedure at the next state. Repeating this procedure until you reach level $H/2$ and union bounding guarantees that with high probability we find a path that leads to either $s^*$ or a state $s \in \mathcal{S}_i$. From here on, one can deterministically follow the linear sequence of states to arrive at $(1)$. The policy that prescribes this path then provides the optimal (unit) value for that MDP, so $\beta = 0$. The total sample complexity of this method is $\Theta(H^{k-1} \times H) = \Theta(H^k)$. Again, note that this argument holds under a generative model as defined in SQM, since we can plug in whatever state we want, and receive as feedback from the generative model the next state sampled from the transition process. Note also that in similar fashion to Theorem 1, this same algorithm can be used to identify a policy achieving optimal value with respect to any initial state in the tree. To convert the policy found by the algorithm into a linear policy, we use the same technique as in Theorem 1.

We now complete the proof of Corollary 2. A basic computation shows that for $k \geq 3$ and sufficiently large $H$, the value (in expectation over $\mathcal{D}$) of any path is at most $\frac{1}{10}$. But the value of the path (in expectation over $\mathcal{D}$) through $s^*$ till $(1)$ is one. Hence any algorithm which can find $\pi$ satisfying $\mathbb{E}_{M \sim \mathcal{D}}[V_M^{s_0}(\pi)] \geq \max_{\pi'} \mathbb{E}_{M \sim \mathcal{D}}[V_M^{s_0}(\pi')] - \frac{1}{4}$ must be able to identify $s^*$ with non-trivial probability. So to show that finding such a $\pi$ requires $\Omega(2^H)$ queries, it is sufficient to show that identifying $s^*$ requires $\Omega(2^H)$ queries. This is done via identical arguments to the ones used to prove Theorem 1.

### A.4 Proof Of Corollary 3

The proof of this corollary is basically identical to the proof of Corollary 1. Briefly, if an algorithm can solve $M_{\text{test}}$ with a number of queries at test time that is strictly fewer than $\Omega(H^k)$, then at training time it must have narrowed the possible locations of $s^*$ to a set whose cardinality is polynomial in $H$. By the proofs of Theorem 1 and Corollary 2, this requires $\Omega(2^H)$ queries during training time.

## B  Upper Bound Proof Details

In this section, we will provide a formal proof of Theorem 2. For ease in presentation, we shall assume throughout that the action space $\mathcal{A}$ for each MDP contains two actions, which we denote $a_1$ and $a_2$. It is easy to extend the proofs to the case when there are many actions. We will often use the notation $s_1$ and $s_2$ to denote the child states of a state $s$ when taking actions $a_1$ and $a_2$ respectively. We shall also use $\pi_M^*$ to denote an optimal policy for MDP $M$. Before proving Theorem 2, we shall state two helpful lemmas. Recall the definition of $\pi^*$ from Strong Proximity (b).

**Lemma 2** *Consider any $\mathcal{D}$ satisfying WIO with $\beta \geq 0$ and Strong Proximity with $\xi_{\text{tr}} = 0$ and any $\alpha, \xi_{\text{r}} \geq 0$, such that the MDPs supporting $\mathcal{D}$ are deterministic. Run Algorithm 1 with $n \geq 1$ samples, and assume at timestep $t$ we are at state $s_t$ such that $\pi^*(s_t) = a_1$. We are guaranteed that the event*

$$\frac{1}{n} \sum_{i=1}^{n} Q_{i,a_1} \geq \frac{1}{n} \sum_{i=1}^{n} Q_{i,a_2} - \alpha - \beta$$

*occurs almost surely. The symmetric statement for $\pi^*(s_t) = a_2$ is also true.*

**Lemma 3** *Consider any $\mathcal{D}$ satisfying WIO with $\beta \geq 0$ and Strong Proximity with $\xi_{\text{tr}} = 0$ and any $\alpha, \xi_{\text{r}} \geq 0$, such that the MDPs supporting $\mathcal{D}$ are deterministic. Run Algorithm 1 with $n = \frac{H^2}{\epsilon^2} \log\left(\frac{2H|\mathcal{A}|}{\delta}\right)$ samples, and assume at timestep $t$ we are at state $s \equiv s_t$. Then the event*

$$\left| \frac{1}{n} \sum_{i=1}^{n} Q_{i,a_1} - \mathbb{E}_{M \sim \mathcal{D}}\left[R_M(s, a_1) + V_M^{s_1}(\pi_M^*)\right] \right| \leq \beta + \frac{\epsilon}{2H}$$

*and*

$$\left| \frac{1}{n} \sum_{i=1}^{n} Q_{i,a_2} - \mathbb{E}_{M \sim \mathcal{D}}\left[R_M(s, a_2) + V_M^{s_2}(\pi_M^*)\right] \right| \leq \beta + \frac{\epsilon}{2H}$$

*occurs with probability at least $1 - \frac{\delta}{H}$.*

The proofs of Lemmas 2 and 3 can be found in Appendix B.2 and Appendix B.3 respectively. With these lemmas in hand, we now turn to the proof of Theorem 2.

### B.1 Proof Of Theorem 2

First observe that by Strong Proximity (b), we have

$$\mathbb{E}_{M \sim \mathcal{D}}\left[V_M^{s_0}(\pi^*)\right] \geq \mathbb{E}_{M \sim \mathcal{D}}\left[\max_{\pi_M} V_M^{s_0}(\pi_M)\right] - \alpha \geq \max_{\pi'} \mathbb{E}_{M \sim \mathcal{D}}\left[V_M^{s_0}(\pi')\right] - \alpha.$$

Hence to prove the theorem, it is sufficient to prove that

$$\mathbb{E}_{M \sim \mathcal{D}}\left[V_M^{s_0}(\pi)\right] \geq \mathbb{E}_{M \sim \mathcal{D}}\left[V_M^{s_0}(\pi^*)\right] - \epsilon - 2\alpha H - 3\beta H,$$

and we shall devote the remainder of the proof to this.

Recall that the policy constructed by Algorithm 1 is represented as a vector of length $H$, which stores an action for each timestep. We use the terminology "algorithm recommends an action at a timestep" to mean that at that timestep, the algorithm stores that action in the policy that it is constructing. Our proof rests on a key claim, which is that while following Algorithm 1, at each timestep the algorithm recommends an action whose suboptimality (in expectation over all MDPs) relative to the other action is at most $\frac{\epsilon}{H} + 2\alpha + 3\beta$. Concretely, assume the algorithm recommends an action $a$ at state $s$ which transports us to $s'$. We claim that with high probability at least $1 - \frac{\delta}{H}$,

$$\mathbb{E}_{M \sim \mathcal{D}}\left[R_M(s, a) + V_M^{s'}(\pi^*)\right] \geq \mathbb{E}_{M \sim \mathcal{D}}\left[V_M^{s}(\pi^*)\right] - \frac{\epsilon}{H} - 2\alpha - 3\beta. \tag{3}$$

First, we argue why this claim in Eq. (3) is sufficient to prove the theorem. Assume this claim to be true, and denote $s_h$ to be the state achieved by the algorithm after $h$ timesteps and $a_h$ to be the action recommended by the algorithm at timestep $h$. Then applying a union bound over $H$ timesteps, with high probability at least $1 - \delta$ we are guaranteed that

$$\mathbb{E}_{M \sim \mathcal{D}}\left[V_M^{s_H}(\pi^*) + \sum_{h=0}^{H-1} R_M(s_h, a_h)\right] = \mathbb{E}_{M \sim \mathcal{D}}\left[R_M(s_{H-1}, a_{H-1}) + V_M^{s_H}(\pi^*)\right] + \mathbb{E}_{M \sim \mathcal{D}}\left[\sum_{h=0}^{H-2} R_M(s_h, a_h)\right]$$

$$\geq \mathbb{E}_{M \sim \mathcal{D}}\left[V_M^{s_{H-1}}(\pi^*) + \sum_{h=0}^{H-2} R_M(s_h, a_h)\right] - \frac{\epsilon}{H} - 2\alpha - 3\beta$$

$$\geq \mathbb{E}_{M \sim \mathcal{D}}\left[V_M^{s_{H-2}}(\pi^*) + \sum_{h=0}^{H-3} R_M(s_h, a_h)\right] - \frac{2\epsilon}{H} - 4\alpha - 6\beta$$

$$\geq \mathbb{E}_{M \sim \mathcal{D}}\left[V_M^{s_0}(\pi^*)\right] - \frac{\epsilon H}{H} - 2\alpha H - 3\beta H$$

$$= \mathbb{E}_{M \sim \mathcal{D}}\left[V_M^{s_0}(\pi^*)\right] - \epsilon - 2\alpha H - 3\beta H.$$

By assumption, the MDPs supporting $\mathcal{D}$ have shared deterministic transitions and a common state-action space $\mathcal{S} \times \mathcal{A}$, and hence the above calculations remain valid. So we have found a sequence of actions $\{a_h\}_{h=0}^{H-1}$, which defines a path through the $\mathcal{S} \times \mathcal{A}$ and enables us to arrive at state $s_H \in \mathcal{S}$

with the above property. Of course $V_M^{s_H}(\pi^*)$ is just trivially zero, since the planning horizon is $H$. So the path $\{a_h\}_{h=0}^{H-1}$ we have found, which defines a deterministic policy denoted by $\pi$, satisfies

$$
\begin{aligned}
\mathbb{E}_{M\sim\mathcal{D}}\left[V_M^{s_0}(\pi)\right] &= \mathbb{E}_{M\sim\mathcal{D}}\left[\sum_{h=0}^{H-1} R_M(s_h, a_h)\right] \\
&= \mathbb{E}_{M\sim\mathcal{D}}\left[V_M^{s_H}(\pi^*) + \sum_{h=0}^{H-1} R_M(s_h, a_h)\right] \\
&\geq \mathbb{E}_{M\sim\mathcal{D}}\left[V_M^{s_0}(\pi^*)\right] - \epsilon - 2\,\alpha\,H - 3\,\beta\,H,
\end{aligned}
$$

which exactly proves the theorem.

Hence it is sufficient to prove the claim in Eq. (3), and we shall devote the remainder to proving this claim. Assume that while running the algorithm we are at some state $s$. Recall the notation that $s_1$ and $s_2$ are the child states of $s$ when taking actions $a_1$ and $a_2$ respectively, and recall $\pi_M^*$ denotes an optimal policy for MDP $M$. By the result of Lemma 3, we have with probability at least $1 - \frac{\delta}{H}$ that

$$
\left|\frac{1}{n}\sum_{i=1}^{n} Q_{i,a_1} - \mathbb{E}_{M\sim\mathcal{D}}\left[R_M(s, a_1) + V_M^{s_1}(\pi_M^*)\right]\right| \leq \beta + \frac{\epsilon}{2H}
$$

$$
\text{and } \left|\frac{1}{n}\sum_{i=1}^{n} Q_{i,a_2} - \mathbb{E}_{M\sim\mathcal{D}}\left[R_M(s, a_2) + V_M^{s_2}(\pi_M^*)\right]\right| \leq \beta + \frac{\epsilon}{2H}. \tag{4}
$$

We condition on this event to verify the claim in Eq. (3).

Now assume WLOG that $\pi^*(s) = a_1$, since the case when $\pi^*(s) = a_2$ is entirely symmetric. By the principle of Bellman optimality, this ensures that for any MDP $M$ we have $R_M(s, a_1) + V_M^{s_1}(\pi^*) = V_M^{s}(\pi^*)$. Furthermore by the result of Lemma 2,

$$
\frac{1}{n}\sum_{i=1}^{n} Q_{i,a_1} \geq \frac{1}{n}\sum_{i=1}^{n} Q_{i,a_2} - \alpha - \beta. \tag{5}
$$

We now consider two cases. For the first case, assume $\frac{1}{n}\sum_{i=1}^{n} Q_{i,a_1} > \frac{1}{n}\sum_{i=1}^{n} Q_{i,a_2}$. Then the algorithm recommends action $a_1$ and transports us to state $s_1$. Recall again by our WLOG assumption that for any MDP $M$, we have $R_M(s, a_1) + V_M^{s_1}(\pi^*) = V_M^{s}(\pi^*)$. So we are guaranteed that

$$
\mathbb{E}\left[R_M(s, a_1) + V_M^{s_1}(\pi^*)\right] = \mathbb{E}\left[V_M^{s}(\pi^*)\right],
$$

and so the claim in Eq. (3) is trivially shown to be true in this case.

For the second case, assume $\frac{1}{n}\sum_{i=1}^{n} Q_{i,a_1} \leq \frac{1}{n}\sum_{i=1}^{n} Q_{i,a_2}$. Then the algorithm will recommend action $a_2$ and transport us to state $s_2$. But note that by the bound Eq. (5), we have

$$
\left|\frac{1}{n}\sum_{i=1}^{n} Q_{i,a_1} - \frac{1}{n}\sum_{i=1}^{n} Q_{i,a_2}\right| \leq \alpha + \beta.
$$

Combining this with the bound Eq. (4), and the triangle inequality, we have

$$
\left|\mathbb{E}_{M\sim\mathcal{D}}\left[R_M(s, a_1) + V_M^{s_1}(\pi_M^*)\right] - \mathbb{E}_{M\sim\mathcal{D}}\left[R_M(s, a_2) + V_M^{s_2}(\pi_M^*)\right]\right| \leq \alpha + 3\,\beta + \frac{\epsilon}{H}.
$$

Hence we have

$$
\begin{aligned}
\mathbb{E}_{M\sim\mathcal{D}}\left[R_M(s, a_2) + V_M^{s_2}(\pi_M^*)\right] &\geq \mathbb{E}_{M\sim\mathcal{D}}\left[R_M(s, a_1) + V_M^{s_1}(\pi_M^*)\right] - \alpha - 3\,\beta - \frac{\epsilon}{H} \\
&\geq \mathbb{E}_{M\sim\mathcal{D}}\left[R_M(s, a_1) + V_M^{s_1}(\pi^*)\right] - \alpha - 3\,\beta - \frac{\epsilon}{H} \\
&= \mathbb{E}_{M\sim\mathcal{D}}\left[V_M^{s}(\pi^*)\right] - \alpha - 3\,\beta - \frac{\epsilon}{H}, \tag{6}
\end{aligned}
$$

where the second inequality follows from the optimality of $\pi_M^*$ for $M$ and the equality follows from our WLOG assumption and the principle of Bellman optimality. Of course Strong Proximity (b) also guarantees that for any MDP $M$ we have $V_M^{s_2}(\pi^*) \geq V_M^{s_2}(\pi_M^*) - \alpha$. We use this to upper bound the LHS of Eq. (6) and obtain

$$\mathbb{E}_{M \sim \mathcal{D}}\left[R_M(s, a_2) + V_M^{s_2}(\pi^*)\right] \geq \mathbb{E}_{M \sim \mathcal{D}}\left[R_M(s, a_2) + V_M^{s_2}(\pi_M^*)\right] - \alpha \geq \mathbb{E}_{M \sim \mathcal{D}}\left[V_M^s(\pi^*)\right] - 2\alpha - 3\beta - \frac{\epsilon}{H}.$$

This exactly demonstrates the claim in Eq. (3), which as argued earlier, is sufficient to complete the proof of the theorem.

## B.2 Proof Of Lemma 2

Let $s \equiv s_t$ and recall that $s_1$ and $s_2$ denote the child states obtained by taking actions $a_1$ and $a_2$ respectively from state $s$. The assumption $\pi^*(s) = a_1$ is WLOG, since the case when $\pi^*(s) = a_2$ is entirely symmetric. Now consider any MDP $M$ and recall $\pi_M^*$ denotes an optimal policy for MDP $M$. There are two cases to consider.

For the first case, assume there exists $\pi_M^*$ such that $\pi_M^*(s) = a_1$. Then,

$$R_M(s, a_1) + V_M^{s_1}(\pi_M^*) = V_M^s(\pi_M^*) \geq R_M(s, a_2) + V_M^{s_2}(\pi_M^*) \geq R_M(s, a_2) + V_M^{s_2}(\pi_M^*) - \alpha,$$

where we used the principle of Bellman optimality.

For the second case, assume there only exists $\pi_M^*$ such that $\pi_M^*(s) = a_2$. Then,

$$\begin{aligned}
R_M(s, a_1) + V_M^{s_1}(\pi_M^*) &\geq R_M(s, a_1) + V_M^{s_1}(\pi^*) \\
&= V_M^s(\pi^*) \\
&\geq V_M^s(\pi_M^*) - \alpha \\
&= R_M(s, a_2) + V_M^{s_2}(\pi_M^*) - \alpha,
\end{aligned}$$

where the first inequality follows from the optimality of $\pi_M^*$ for $M$, the equalities follow from the principle of Bellman optimality as well as the WLOG assumption that $\pi^*(s) = a_1$, and the second inequality follows from Strong Proximity (b).

So in either case we are guaranteed that

$$R_M(s, a_1) + V_M^{s_1}(\pi_M^*) \geq R_M(s, a_2) + V_M^{s_2}(\pi_M^*) - \alpha.$$

Recall that for action $a$ leading to state $s'$ from state $s$, we have $Q_{i,a} = R_{M_i}(s, a) + \widehat{V}_{M_i}^{s'}$ by definition. We are then guaranteed by WIO that

$$R_{M_i}(s, a) + V_{M_i}^{s'}(\pi_{M_i}^*) \geq Q_{i,a} \geq R_{M_i}(s, a) + V_{M_i}^{s'}(\pi_{M_i}^*) - \beta.$$

Hence for any $M_i$ we must have

$$\begin{aligned}
Q_{i,a_1} &= R_{M_i}(s, a_1) + \widehat{V}_{M_i}^{s_1} \\
&\geq R_{M_i}(s, a_1) + V_{M_i}^{s_1}(\pi_{M_i}^*) - \beta \\
&\geq R_{M_i}(s, a_2) + V_M^{s_2}(\pi_{M_i}^*) - \alpha - \beta \\
&\geq R_{M_i}(s, a_2) + \widehat{V}_{M_i}^{s_2} - \alpha - \beta \\
&= Q_{i,a_2} - \alpha - \beta.
\end{aligned}$$

Averaging each side of the above inequality over $n$ completes the proof.

## B.3 Proof Of Lemma 3

Note that for action $a$ leading to state $s'$ from state $s$, we have $Q_{i,a} = R_{M_i}(s, a) + \widehat{V}_{M_i}^{s'}$ by definition. We are then guaranteed by WIO that

$$R_{M_i}(s, a) + V_{M_i}^{s'}(\pi_{M_i}^*) \geq Q_{i,a} \geq R_{M_i}(s, a) + V_{M_i}^{s'}(\pi_{M_i}^*) - \beta.$$

Hence we must have

$$\frac{1}{n}\sum_{i=1}^{n}(R_{M_i}(s,a) + V_{M_i}^{s'}(\pi_{M_i}^*)) \geq \frac{1}{n}\sum_{i=1}^{n}Q_{i,a} \geq \frac{1}{n}\sum_{i=1}^{n}(R_{M_i}(s,a) + V_{M_i}^{s'}(\pi_{M_i}^*)) - \beta.$$

By Hoeffding's bound and our choice of $n$, we are guaranteed that with probability at least $1 - \frac{\delta}{H}$ that

$$\left|\frac{1}{n}\sum_{i=1}^{n}(R_{M_i}(s,a_1) + V_{M_i}^{s_1}(\pi_{M_i}^*)) - \mathbb{E}_{M\sim\mathcal{D}}\left[R_M(s,a_1) + V_M^{s_1}(\pi_M^*)\right]\right| \leq \frac{\epsilon}{2H}$$

$$\text{and } \left|\frac{1}{n}\sum_{i=1}^{n}(R_{M_i}(s,a_2) + V_{M_i}^{s_2}(\pi_{M_i}^*)) - \mathbb{E}_{M\sim\mathcal{D}}\left[R_M(s,a_2) + V_M^{s_2}(\pi_M^*)\right]\right| \leq \frac{\epsilon}{2H}.$$

So for action $a$ leading to state $s'$ from state $s$, we can combine the previous two equations via the triangle inequality to obtain

$$\left|\frac{1}{n}\sum_{i=1}^{n}Q_{i,a} - \mathbb{E}_{M\sim\mathcal{D}}\left[R_M(s,a) + V_M^{s'}(\pi_M^*)\right]\right|$$

$$\leq \left|\frac{1}{n}\sum_{i=1}^{n}(R_{M_i}(s,a) + V_{M_i}^{s'}(\pi_{M_i}^*)) - \frac{1}{n}\sum_{i=1}^{n}Q_{i,a}\right|$$

$$+ \left|\frac{1}{n}\sum_{i=1}^{n}(R_{M_i}(s,a) + V_{M_i}^{s'}(\pi_{M_i}^*)) - \mathbb{E}_{M\sim\mathcal{D}}\left[R_M(s,a) + V_M^{s'}(\pi_M^*)\right]\right|$$

$$\leq \beta + \frac{\epsilon}{2H}.$$

This completes the proof.

## C  Near Tightness Of Theorem 2

As discussed in Section 4.2, the $\alpha, \beta$ terms in the error bound of Theorem 2 scale linearly with $H$. So when either $\alpha$ or $\beta$ is $\Omega(\frac{1}{H})$, then our bound becomes vacuous. It is natural to question whether this unfortunate scaling is due to a possible suboptimality of Algorithm 1 or some looseness in our analysis. The following result provides a (partial) answer to this question.

**Proposition 1** *Let $n$ be the total query cost that any algorithm is allowed to use under WQM. For any $\beta \geq 0$, there exists $\mathcal{D}$ satisfying WIO with $\beta$ and Strong Proximity with $\xi_r = \xi_{tr} = \alpha = 0$, such that the MDPs supporting $\mathcal{D}$ are deterministic and the following holds. Any (possibly randomized) algorithm will output (with probability at least $\frac{1}{2}$) a policy $\pi$ satisfying*

$$\mathbb{E}_{M\sim\mathcal{D}}[V_M^{s_0}(\pi)] \leq \max_{\pi'}\mathbb{E}_{M\sim\mathcal{D}}[V_M^{s_0}(\pi')] - \frac{\beta H}{\log(50n)}.$$

This result demonstrates that the dependency on $\beta$ given in the result of Theorem 2 is tight to within a logarithmic factor in $H$, and cannot be improved beyond this logarithmic factor by a better algorithm or sharper analysis. It remains unclear whether the dependence on $\alpha$ is tight. Isolating this is more difficult, since in any construction $\alpha$ and $\beta$ become inherently intertwined, and it is unclear to us how to separate the dependencies on these two distinct parameters. We believe this is an interesting direction for future work. Intuitively the dependency on $\alpha$ seems tight — roughly speaking it seems natural that if we are planning over $H$ timesteps, while using the structure induced by $\pi^*$ to guide our actions, then the suboptimality of $\alpha$ at each timestep leads to a total error blowup of $\alpha H$. Nevertheless, a formal proof (or otherwise) would be an interesting development. We now turn to proving Proposition 1.

**Proof.** Fix any $\beta \geq 0$. It is sufficient to construct a single MDP $M$ with deterministic transitions, and construct an oracle $\widehat{V}$ satisfying WIO with parameter $\beta$, and show that any algorithm will output a policy $\pi$ that satisfies

$$V_M^{s_0}(\pi) \leq \max_{\pi'}V_M^{s_0}(\pi) - \frac{\beta H}{\log(50n)},$$

with probability at least $\frac{1}{2}$. This is because we can define $\mathcal{D}$ to be the point mass on $M$, so that Strong Proximity (a) and (b) are satisfied trivially with $\xi_{\mathrm{r}} = \xi_{\mathrm{tr}} = \alpha = 0$. We will define $M$ as a binary tree, where the states are nodes in the tree, and the (deterministic) actions are described by the edges connecting nodes, which immediately ensures that $M$ is deterministic. We will construct the following MDP $M$ by chunking the levels of the tree into blocks. Each block has length $\log(50n)$. To facilitate the definition of the reward function used to define $M$, we will first assign each state its optimal value, i.e. the value that one could get by following the optimal policy from that state. This will naturally allow us to later define rewards. We will assign these optimal values of the states in a sequential fashion, by considering each subtree rooted at some state on level $h_k = k \log(50n)$, where $k$ is a nonnegative integer.

Start by considering level $h_0 = 0$. Then on level $h_1 = \log(50n)$, pick a single state uniformly at random, denote it $s_{h_1, s_0}$, and assign it value that satisfies $V^{s_{h_1, s_0}}(\pi^*) = V^{s_0}(\pi^*)$. We will call this state the special state for level $h_1$. Let all other states $s$ on level $h_1$ have value $V^s(\pi^*) = V^{s_0}(\pi^*) - \beta$.

Recursively define the values of the leaves below this level in an identical fashion. To be more concrete, do the following procedure for each state $s$ on level $h_1 = \log(50n)$. Consider all the states in the subtree of $s$ that lie on level $h_2 = 2\log(50n)$. Pick a single one of these states uniformly at random, denote it $s_{h_2, s}$, and assign it value that satisfies

$$V^{s_{h_2, s}}(\pi^*) = V^s(\pi^*).$$

We will call this state one of the special states for level $h_2$. And for all other states $s' \neq s_{h_2, s}$ in the subtree of $s$ that lie on level $h_2 = 2\log(50n)$, assign them each value that satisfies

$$V^{s'}(\pi^*) = V^s(\pi^*) - \beta.$$

In this fashion, we can assign values for every state that lies on a level $k\log(50n)$ where $k$ is a nonnegative integer. We assume without loss of generality that $\log(50n)$ evenly divides $H$. Of course, this immediately defines the values for all states in the tree. To actually define rewards which satisfies the structure induced by the values, fix $V_M^{s_0}(\pi^*)$ to be any number, and then simply assign rewards to the leaves of the tree which agree with the assigned values of those leaves. The reward is zero for all states in the tree that are not leaves.

We now define the oracle $\widehat{V}$. For any state $s$ in the levels $h$ satisfying $h_0 \leq h \leq h_1$, let $\widehat{V}(s) = V^{s_0}(\pi^*) - \beta$. Similarly, consider any state $s$ on level $h_1$, and now consider any state $s' \neq s$ in the subtree rooted at $s$ such that $\mathrm{level}(s') \leq h_2$. Then let $\widehat{V}(s') = V^s(\pi^*) - \beta$. Recursively repeat this until we have defined $\widehat{V}$ for each state in the tree. By definition, we have defined $\widehat{V}$ so that it satisfies WIO with parameter $\beta$.

With this construction in hand, the proof is now straightforward to complete. It is sufficient to prove this in the case when an algorithm returns a deterministic policy (or path), since any stochastic policy is a randomization over deterministic paths. We claim that with probability at least $\frac{1}{2}$, the path outputted by the algorithm never intersects *any* of the special states in the *entire* tree. It is sufficient to prove this, because this implies that the path loses $\beta$ value for a total of $\frac{H}{\log(50n)}$ times, implying that its value is at most

$$V_M^{s_0}(\pi^*) - \frac{\beta H}{\log(50n)}.$$

By symmetry, it is clear that any algorithm which can identify even a single special state anywhere in the tree with non-trivial probability, can be used to identify the special state on level $h_1$. Hence, it is sufficient to prove that there is no algorithm which can identify the special state on level $h_1$ with non-trivial probability. Strengthen the query model (slightly) so that querying a state will return whether it is the special state on level $h_1$. But now, observe that we are only allowed to query $n$ states total. And to identify the special state on level $h_1$, the algorithm must query most of the states on level $h_1$, which is a total of

$$2^{\log(50n)} = 50n$$

states. This implies that it cannot identify the special state on level $h_1$ with probability at least $\frac{1}{2}$. As discussed above, this is sufficient to complete the proof. $\square$

# D    Auxiliary Proof Details

In this section, we prove Lemma 1, relying heavily on the treatment given in [25]. Note that in our setting, we only use this result in the context of our lower bounds, and all our lower bounds have finite state-action spaces. So it is sufficient to prove the result assuming that the state-action space is finite.

Let $\tau$ denote a trajectory, and let $\mathbb{P}_1^\pi(\tau), \mathbb{P}_2^\pi(\tau)$ denote the probabilities of taking $\tau$ in MDPs $M_1, M_2$ respectively. Let $R_1(\tau), R_2(\tau)$ denote the rewards obtained by following $\tau$ in MDPs $M_1, M_2$ respectively. The penultimate step of the proof of Lemma 4.3 in [25] shows that

$$\sum_\tau |\mathbb{P}_1^\pi(\tau) - \mathbb{P}_2^\pi(\tau)| \leq \xi_{\mathrm{tr}}\, H. \tag{7}$$

The definition of $\xi_{\mathrm{r}}$ and a straightforward application of the triangle inequality reveals that

$$|R_1(\tau) - R_2(\tau)| \leq \sum_{t=0,(s_t,a_t)\in\tau}^{H-1} |R_{M_1}(s_t, a_t) - R_{M_2}(s_t, a_t)| \leq \xi_{\mathrm{r}}\, H. \tag{8}$$

Again using the triangle inequality, we obtain that

$$
\begin{aligned}
\left|V_{M_1}^{s_0}(\pi) - V_{M_2}^{s_0}(\pi)\right| &= \left|\sum_\tau \left(\mathbb{P}_1^\pi(\tau)R_1(\tau) - \mathbb{P}_2^\pi(\tau)R_2(\tau)\right)\right| \\
&\leq \left|\sum_\tau \left(\mathbb{P}_1^\pi(\tau)R_1(\tau) - \mathbb{P}_1^\pi(\tau)R_2(\tau)\right)\right| + \left|\sum_\tau \left(\mathbb{P}_1^\pi(\tau)R_2(\tau) - \mathbb{P}_2^\pi(\tau)R_2(\tau)\right)\right| \\
&\leq \sum_\tau \mathbb{P}_1^\pi(\tau)\,|R_1(\tau) - R_2(\tau)| + \left|\sum_\tau \left(\mathbb{P}_1^\pi(\tau)R_2(\tau) - \mathbb{P}_2^\pi(\tau)R_2(\tau)\right)\right| \\
&\leq \xi_{\mathrm{r}}\, H + \left|\sum_\tau \left(\mathbb{P}_1^\pi(\tau)R_2(\tau) - \mathbb{P}_2^\pi(\tau)R_2(\tau)\right)\right|,
\end{aligned}
$$

where we used the result of Eq. (8). Furthermore,

$$
\begin{aligned}
\left|V_{M_1}^{s_0}(\pi) - V_{M_2}^{s_0}(\pi)\right| &\leq \xi_{\mathrm{r}}\, H + \left|\sum_\tau \left(\mathbb{P}_1^\pi(\tau)R_2(\tau) - \mathbb{P}_2^\pi(\tau)R_2(\tau)\right)\right| \\
&\leq \xi_{\mathrm{r}}\, H + \sum_\tau |\mathbb{P}_1^\pi(\tau) - \mathbb{P}_2^\pi(\tau)|\,|R_2(\tau)| \\
&\leq \xi_{\mathrm{r}}\, H + \sum_\tau |\mathbb{P}_1^\pi(\tau) - \mathbb{P}_2^\pi(\tau)| \\
&\leq \xi_{\mathrm{r}}\, H + \xi_{\mathrm{tr}}\, H,
\end{aligned}
$$

where we used the result of Eq. (7) and also our assumption that the reward of any trajectory is always upper bounded by one. This completes the proof.