# OpenReview forum: "When Is Generalizable Reinforcement Learning Tractable?"
_NeurIPS.cc/2021/Conference — NeurIPS 2021 Poster_

### Official Review · Reviewer_q4SK · 2021-07-16

**Rating:** 7
**Confidence:** 2

**Summary:**

This paper studies theoretically the challenge of learning policies that generalize and establish bounds showing (1) in the worst case tractable generalization is impossible; but (2) can be efficiently learnt in cases that meet the conditions proposed for strong proximity.

**Limitations And Societal Impact:**

The paper gives a refreshingly open-minded discussion of limitations and potential for long-term negative societal impacts. In particular, I appreciate the authors efforts to use this as a way to further clarify the limits of these findings and motivate future work.

**Main Review:**

The paper is timely given significant interest across the community in enabling reinforcement learning to generalize beyond the training environment and provides novel insights into when the task is and is not tractable. The clarity of the goals of this paper could be improved by formally establishing earlier on what conditions the authors have chosen for classifying a solution as tractable.

The bounds are limited in relevance to environments that share state-action spaces, but this assumption is clearly stated on page 2 of the paper, discussed in the limitations of the work and still provides novel insight.  The clarity of these assumptions could be improved by using a concrete example (e.g. a gridworld) throughout as a narrative tool (Please note I am not implying empirical results are needed, only that a concrete example can help readers quickly assess the relevance of theory to their applications).

Similarly, on page 5, the definition of strong individual optimization assumes there is an efficient algorithm for learning a policy to a single MDP without linking interested readers to existing published methods that have been proven to meet these requirements. If there are none, then this is an interesting point to highlight for others to pursue in future work. Negative examples of well established methods that do not meet these requirements could also be useful to help readers connect theory to practice.

The lower bound demonstrates worst case under weak proximity, strong query model and strong individual optimization but it remains unclear to me what happens under other combinations? In particular, as the upper bound is proven for strong proximity even with a weak query model and weak individual optimization, can it be implied that weak proximity alone makes generalization intractable?

**Time Spent Reviewing:**

4

---

> ### Author Response · Authors · 2021-08-09
> **Thank you**
>
> Thank you for your review and your kind words about our paper. Below, we address your questions. Please let us know if anything remains unclear.
>
> **0. The clarity of the goals of this paper could be improved by formally establishing earlier on what conditions the authors have chosen for classifying a solution as tractable.**
>
> We stated what it means for a solution to be tractable, when we described the WQM and SQM query models, specifically when we said "the goal is to solve Eqs. (1) and (2) with a total query cost that is at most polynomial in...". This requirement follows a long line of RL theory work for the function approximation RL setting (see, for instance, the papers [A], [B], [C], [D], [E], [F]). If you feel that this is not clear enough, then please let us know, and we will emphasize it more in the camera ready.
>
> **1. The bounds are limited in relevance to environments that share state-action spaces, but this assumption is clearly stated on page 2 of the paper, discussed in the limitations of the work and still provides novel insight.**
>
> We agree that the assumption that the MDPs share a state-action space limits the applicability of our *upper* bound. However, the reason we studied the shared state-action space setting, is because we proved a *lower* bound for this setting in the case of Weak Proximity, so it is natural to study this same setting when considering Strong Proximity to make the results more comparable. Indeed, we highlight that the shared state-action space assumption *strengthens* our lower bound, since a lower bound which applies to this special case of course applies to the more general realistic case of when the MDPs have different state spaces. Finally, as we said in the Future Work subsection of our Discussion, we remark that if one is given a feature map which maps the state spaces of different MDPs to the same feature space, then our upper bound insights can conceivably be applied.
>
> **2. The clarity of these assumptions could be improved by using a concrete example (e.g. a gridworld) throughout as a narrative tool.**
>
> We did mention a concrete robotic goal reaching setting (see the paper [I]) to clarify Condition 1. This is described in Lines 172-177, right after the statement of Condition 1. However, we agree that it is useful to use this same example to help clarify the other subsequent Conditions/Properties. In the camera ready, we can expand upon this example to clarify the subsequent Conditions/Properties. For instance, the paper [I] empirically shows that individual MDPs in their family of tasks can be efficiently solved via a variant of policy gradient, and we can cite this when motivating our individual optimization properties. Space permitting, we can also include an additional example such as GridWorld.
>
> **3. Similarly, on page 5, the definition of strong individual optimization assumes there is an efficient algorithm for learning a policy to a single MDP without linking interested readers to existing published methods that have been proven to meet these requirements. If there are none, then this is an interesting point to highlight for others to pursue in future work. Negative examples of well established methods that do not meet these requirements could also be useful to help readers connect theory to practice.**
>
> We stated the SIO property with respect to a generic efficient algorithm, since it is well known that MDPs with different structures can require different types of algorithms to solve efficiently. This has been noted, for instance, in the papers [A], [B], [C], [D], [E], [F]. Since MDPs with a variety of different structures can satisfy SIO, it is not clear a priori what is the right algorithm to use for that MDP (at least from a strictly theoretical perspective), and so we did not mention a specific algorithm in the formal statement of SIO. We can clarify this in the camera ready.
>
> However, we note that for the specific MDP family construction used in our lower bound, the algorithm we provide to satisfy the SIO property is well known and classical. Indeed, it is a greedy version of Monte Carlo Tree Search, which has been extremely popular in RL (see, for instance, the papers [G] and [H]). We are happy to mention this in the camera ready version.
>
> **4. The lower bound demonstrates worst case under weak proximity, strong query model and strong individual optimization but it remains unclear to me what happens under other combinations? In particular, as the upper bound is proven for strong proximity even with a weak query model and weak individual optimization, can it be implied that weak proximity alone makes generalization intractable?**
>
> Note that any other combination follows as a corollary of our main results. For instance, our lower bound immediately holds with Weak Proximity, WIO, and WQM, since we proved it holds with Weak Proximity and the stronger SQM and stronger SIO. Similarly, our upper bound immediately holds with Strong Proximity, SQM and SIO, since we proved that it holds with Strong Proximity and the weaker WIO and WQM.
>
> It is not very precise to say that Weak Proximity alone makes generalization intractable, since one needs to state the query model and the complexity required to solve individual MDPs, in order to fully specify the problem. To see this, consider the following extreme case. In the context of our Weak Proximity lower bound, imagine that the agent had access to an individual optimization oracle that could query every state in an MDP and return *all* of the optimal policies in the MDP via $\mathcal{O}(1)$ complexity. In such a case, our lower bound would no longer apply and generalization would be trivial. But of course, such an oracle is extremely unreasonable, and could not be implemented, especially since each MDP in our lower bound has $\mathcal{O}(2^H)$ different policies achieving optimal value.
>
> **Papers Cited:**
>
> [A] Jin, Yang, Wang, Jordan. Provably Efficient Reinforcement Learning with Linear Function Approximation, 2019.
>
> [B] Weisz, Amortila, Janzer, Abbasi-Yadkori, Jiang, Szepesvári. On Query-efficient Planning in MDPs under Linear Realizability of the Optimal State-value Function, 2021.
>
> [C] Lattimore, Szepesvári, Weisz. Learning with good feature representations in bandits and in rl with a generative model, 2020.
>
> [D] Malik, Pacchiano, Srinivasan, Li. Sample Efficient Reinforcement Learning In Continuous State Spaces: A Perspective Beyond Linearity, 2021.
>
> [E] Wang, Salakhutdinov, Yang. Reinforcement Learning with General Value Function Approximation: Provably Efficient Approach via Bounded Eluder Dimension, 2020.
>
> [F] Du, Lee, Mahajan, Wang. Agnostic Q-learning with Function Approximation in Deterministic Systems: Near-Optimal Bounds on Approximation Error and Sample Complexity, 2020.
>
> [G] Kocsis, Szepesvari. Bandit Based Monte Carlo Planning, 2006.
>
> [H] Guo, Singh, Lee, Lewis, Wang. Deep Learning for Real-Time Atari Game Play Using Offline Monte-Carlo Tree Search Planning, 2014.
>
> [I] Yu, Quillen, He, Julian, Hausman, Finn, Levine. Meta-world: A benchmark and evaluation for multi-task and meta reinforcement learning, 2019.

---

> > ### Comment · Reviewer_q4SK · 2021-08-16
> > **Reply To Authors**
> >
> > Thank you for the detailed response, it has helped improve my understanding and appreciation for this work. A few quick responses:
> >
> > **Regarding Response 0**
> > I still think it would be beneficial to specify this definition of efficiency in the introduction and to reference these papers in support of this being a sufficient definition of tractable. As notation and equations 1 and 2 will not have been formally introduced at this stage this may require a longer form explanation, but it will immediately establish for all readers the goal of the paper and significance of the contribution.
> >
> > **Regarding Response 2**
> > The use of the robotic goal reaching setting in Lines 172-177 is a great example of the benefit of a concrete example. If this can be used similarly throughout, then I see no need for an additional gridworld setting. Any single concrete setting that can be used throughout will be a great help in making the results more accessible.

---

> > > ### Author Response · Authors · 2021-08-16
> > > **We will do as you have recommended on these 2 points**
> > >
> > > Thank you for your follow up and for carefully reading our response.
> > >
> > > Re Response 0: Yes, we agree that discussing the definition of tractable will be useful in the introduction, so that readers understand our goals and contributions early on. We will do this in the camera ready. We will be sure to also refer to the papers we cited in our response above.
> > >
> > > Re Response 2: Sure, we will gladly expand on the robotic goal reaching setting, and use it regularly throughout the paper to help explain the Conditions and Properties.
> > >
> > > Please let us know if you have any other comments or suggestions.

---

### Official Review · Reviewer_q3V1 · 2021-07-16

**Rating:** 7
**Confidence:** 3

**Summary:**

This paper studies the sample complexity of reinforcement learning in families of MDPs that share some notion of structure. The paper considers different notions of structure and sample complexity, and finds that even a relatively strong notion of structure where all MDPs in the family share an optimal linear policy and similar reward/transition functions, finding this shared optimal policy can require time exponential in the horizon of the problem. The paper also presents a stronger notion of structure under which finding a policy that attains near-optimal returns in all environments is tractable.


**Limitations And Societal Impact:**

Theoretical work, N/A

**Main Review:**

Strengths

- Clarity: the paper is very well-written and does a good job at motivating the problem setting. I found the proofs for the most part very clear and appreciated the intuition given in the main body
.
- Novelty: While prior work has studied the sample complexity of multi-environment RL, I am not aware of any existing literature that explicitly studies the type of shared structure needed for generalization to new environments to be tractable.

- Significance and relevance to the community: a lot of the empirical meta-RL literature implicitly assumes some notion of shared structure between tasks but doesn’t quantify this explicitly. This paper seems like an interesting first step towards characterizing what kinds of shared structure are necessary for policies learned in one environment to generalize to others.

- Technical correctness: I didn’t spot any errors in the proofs included in the appendix.

Weaknesses:

- The main “trick” to get lower bounds involves turning the problem of finding a shared optimal policy into a brute-force search over a set of states of size exponential in the horizon. While this is a nice proof of existence, most of the work on sample-efficient RL I’m aware of tends to include the size of the state space in the sample complexity. As a result, the exponential sample size requirement looks like it ends up becoming linear in the size of the state space, which doesn’t seem to be a particularly concerning lower bound. It seems like the increased sample complexity over the single-environment setting would then come from the power of the oracle being used to identify the policies. Would the exponential increase in sample complexity still hold if the query model assumes complexity linear in the size of the state space to compute the value of a state?

- The Strong Proximity assumption is indeed very strong and severely limits the generality of Theorem 2. In most settings of interest we cannot say that a deterministic optimal policy is shared by all of the environments in all states, particularly when the rewards are permitted to vary .

- The optimization oracles (even the weaker “WIO”) seem very strong. As mentioned previously, it seems like a fairly large assumption to assume that the value oracle can output the value of a state in sample complexity polynomial in the horizon and constant in the size of the state space.

Minor points:

I think a visualization of the MDP structure used for corollary 2 would be useful to improve clarity, as it took me a couple of reads to fully understand how all of the states were connected.

**Time Spent Reviewing:**

5.5

---

> ### Author Response · Authors · 2021-08-09
> **Thank you**
>
> Thank you for your review and your kind words about our paper. Below, we address your questions. Please let us know if anything remains unclear.
>
> **1. The main “trick” to get lower bounds involves turning the problem of finding a shared optimal policy into a brute-force search over a set of states of size exponential in the horizon. While this is a nice proof of existence, most of the work on sample-efficient RL I’m aware of tends to include the size of the state space in the sample complexity. As a result, the exponential sample size requirement looks like it ends up becoming linear in the size of the state space, which doesn’t seem to be a particularly concerning lower bound. It seems like the increased sample complexity over the single-environment setting would then come from the power of the oracle being used to identify the policies. Would the exponential increase in sample complexity still hold if the query model assumes complexity linear in the size of the state space to compute the value of a state?**
>
> In the *tabular* RL setting, it is natural and necessary for the sample complexity to scale with the size of the state space. However, our paper studies the *function approximation* RL setting, where each state is associated with a feature, so that the state space is a subset of $\mathbb{R}^d$. In this setting, the number of states is in general very large (perhaps even uncountable), and one desires a sample complexity bound that is independent of the cardinality of the state space, and instead scales polynomially with the horizon $H$, the cardinality of the action space (which is finite and small in our setting as well as many applications such as video games), and the dimensionality $d$ of the features. This is very standard, and there is a very long line of RL theory work which studies when this is possible (see, for instance, the papers [A], [B], [C], [D], [E]).
>
> We noticed that when presenting our query models, we forgot to mention that the sample complexity should depend polynomially on the dimension $d$, and this is probably what led to the confusion. Thank you for bringing this to our attention! We apologize for our accidental omission, and we are happy to clarify this in the camera ready.
>
> We emphasize that this in no way affects the correctness of our results. In our lower bound construction, we made sure the dimension $d$ is polynomial in $H$ (**indeed this is one of the challenges of the proof**), and so the lower bound remains correct. And the sample complexity of our upper bound is independent of $d$ anyways, and so is unaffected.
>
> With these considerations, note that in our function approximation setting, it does not make sense to consider an individual optimization oracle whose complexity is linear in the size of the state space, since the size of the state space could be infinite. This is why we said that the SIO and WIO oracles operate in complexity that is polynomial in $H$ and $\vert \mathcal{A} \vert$, and we will certainly additionally mention in the camera ready that their complexity is also polynomial in the dimension $d$.
>
> **2. The Strong Proximity assumption is indeed very strong and severely limits the generality of Theorem 2. In most settings of interest we cannot say that a deterministic optimal policy is shared by all of the environments in all states, particularly when the rewards are permitted to vary.**
>
> We agree that our notion of Strong Proximity is rather strong. As discussed in our Limitations section, our motivation for this was simply because we already proved a lower bound for Weak Proximity, and we wanted to take an initial step at identifying a sufficient condition for generalization. Note that our characterization of Strong Proximity shows that a condition which is both necessary and sufficient for generalization, lies between Weak and Strong Proximity --- at least if we do not make even stronger assumptions than SIO on the tractability of individual MDPs, which would arguably be unreasonable. Identifying this simultaneously necessary and sufficient condition, is a key direction for future work, and we believe that our own work lays the foundation for this future work.
>
> We would also like to mention that if we were a priori given a feature mapping that maps MDPs with distinct state spaces to the same low dimensional feature space, then it is more reasonable that Strong Proximity can hold and our upper bound might apply. For instance, in the CoinRun generalization benchmark, even though the different MDPs have disjoint high dimensional state spaces, the minimal representation required to generalize is very small (it only consists of the agent's distance from the forthcoming obstacle, so the agent knows whether to jump or not). So when given a lower dimensional feature map, it is more reasonable to expect that Strong Proximity holds on the shared feature space. But of course, learning such a feature map poses its own set of challenges, and requires future investigation.
>
> **3. The optimization oracles (even the weaker “WIO”) seem very strong. As mentioned previously, it seems like a fairly large assumption to assume that the value oracle can output the value of a state in sample complexity polynomial in the horizon and constant in the size of the state space.**
>
> Please see our response to your above comment, about how in our function approximation setting, one desires a sample complexity bound that is independent of the cardinality of the state space (which can be infinite). We will be sure to additionally mention that complexity of SIO/WIO is polynomial in the dimension $d$. Thank you again for bringing this to our attention, although we emphasize again that this in no way affects the correctness of our results.
>
> We agree that SIO is strong, and this only makes our lower bounds stronger. We do not, however, believe that WIO is very strong. WIO does not even return an optimal policy for an individual MDP, and only returns a scalar value. Note that in many applications such as video games (for example in the CoinRun generalization benchmark), the reward function is the indicator for whether the agent wins the game, and this reward signal is only received once at the end of the game. So the optimal value achievable from a state is also binary and may be very easily computable, without having to implement an entire RL algorithm for an individual MDP.
>
> If an oracle cannot return even the basic information returned by WIO, then we do not think it is possible to guarantee provable generalization, even when the Strong Proximity condition holds perfectly with $\alpha = 0$! We did provide some formal evidence for this in our paper. In Appendix C of our paper, we proved a lower bound, where we constructed a distribution $\mathcal{D}$ satisfying Strong Proximity with $\alpha = 0$ such that the following holds. When given access to WIO with suboptimality $\beta$, any algorithm that uses only polynomial sample complexity will suffer $\Omega(\beta \frac{H}{\log(H)})$ generalization error. See Appendix C for the formal statement. This implies that if the WIO oracle is corrupted or has suboptimality larger than $\Omega(\frac{\log(H)}{H})$, then provable generalization is impossible, even when the Strong Proximity condition holds perfectly! So, our notion of WIO (or something analogous) is indeed necessary, at least in long horizon problems, because large $H$ implies $\frac{\log(H)}{H} \approx 0$.
>
> **4. I think a visualization of the MDP structure used for corollary 2 would be useful to improve clarity, as it took me a couple of reads to fully understand how all of the states were connected.**
>
> The visualization for Corollary 2 is provided in Figure 2, located on page 18 of the complete paper (including the Appendix). If you feel that it is not sufficient or clear, we are happy to update it for the camera ready version.
>
> **Papers Cited:**
>
> [A] Du, Kakade, Wang, Yang. Is a Good Representation Sufficient for Sample Efficient Reinforcement Learning? 2019.
>
> [B] Wang, Foster, Kakade. What are the Statistical Limits of Offline RL with Linear Function Approximation? 2021.
>
> [C] Weisz, Amortila, Szepesvári. Exponential Lower Bounds for Planning in MDPs With Linearly-Realizable Optimal Action-Value Functions, 2021.
>
> [D] Malik, Pacchiano, Srinivasan, Li. Sample Efficient Reinforcement Learning In Continuous State Spaces: A Perspective Beyond Linearity, 2021.
>
> [E] Wang, Salakhutdinov, Yang. Reinforcement Learning with General Value Function Approximation: Provably Efficient Approach via Bounded Eluder Dimension, 2020.

---

> > ### Comment · Reviewer_q3V1 · 2021-08-28
> > **Thanks**
> >
> > Thank you for clarifying my concerns, particularly with regard to computing sample complexity w.r.t. the feature dimension rather than the size of the state space. I think the paper will benefit from discussing the relevance of feature dimension to the notion of sample complexity used in the paper. I remain happy to recommend this paper for acceptance.

---

> > > ### Author Response · Authors · 2021-08-30
> > > **Thank you**
> > >
> > > Thank you for reading our rebuttal carefully. We apologize again for our accidental omission, but luckily it is a very easy fix. This will absolutely be corrected in the camera ready version.

---

### Official Review · Reviewer_WHS3 · 2021-07-17

**Rating:** 7
**Confidence:** 4

**Summary:**

The paper provides evidence, in the context of proving generalization in RL, for moving away from classical definitions of proximity typically derived from the simulation lemma (closeness in rewards and transitions across MDPs), and instead, adopting a more fine-grained notion called “strong proximity” which constraints the family of MDPs to have a shared policy with near-optimal value at any state.

The paper consists of two key results or bounds, assuming a family of MDPs with the same state-action space, deterministic transitions, and the usual simulation-lemma proximity parameters for reward and transitions:

(a) An exponential lower bound on efficiency that uses the a traditional notion of “weak proximity” (assuming a shared policy with near-optimal value from the start state) along with strong assumptions on the solvability of each individual MDP in the family (“Strong Individual Optimization” property) as well as the availability of a generative model which can be queried at any arbitrary state and action (in contrast to episodic RL). This lower bound demonstrates that weak proximity does not admit efficient generalization in the worst case despite these strong assumptions.

(b) A polynomial upper bound on efficiency with the proposed strong proximity property with weaker assumptions on the solvability of each individual MDP (“Weak Individual Optimization”) and relaxing the assumption of availability of a generative model (episodic RL setting). This result, with the proposed algorithm, demonstrates a sufficient condition for efficient generalization.

Both bounds are proven for the “Average Performance Setting” where the goal is to find an optimal policy over a distribution of MDPs and the “Meta Reinforcement Learning Setting” where the goal is to maximize performance on a test MDP sampled from the same distribution as train and with some fine-tuning.


**Limitations And Societal Impact:**

The authors have adequately addressed limitations and potential negative societal impacts in the Discussion section.

**Main Review:**

Overall, the paper advances our understanding of the right set of proximity measures that are sufficient for guaranteeing generalization in RL. The lower bound on efficiency with strong assumptions provides a surprising negative result for generalization and the upper bound with weak assumptions paves a path forward given the proposed strong proximity condition. The paper is theoretically sound, very clear in terms of definitions, assumptions, motivations and provides well-written proofs that are accessible to a wider audience. The only weaknesses lie in the inherent assumptions made for each theoretical result in the paper. My rating for this paper is guided by these contributions.

Below are specific strengths and weaknesses that were considered for my rating.

## Strengths
1. As stated earlier, the first main result (Theorem 1) which is an exponential lower bound on efficiency with the strong assumptions of individual solvability (SIO) and the availability of a generative model (Strong Query Model) is a surprising negative result. The proof is well-written and constructs a worst-case scenario of a family of MDPs where at least an exponential number of queries need to be made to even reach the “hidden state” from which the shared optimal policy can be obtained. The binary-tree constructions of the MDP family and the featurization of the state-action space that does not leak the location of the “hidden state” (inspired from prior work) lead to an elegant proof of Theorem 1.

2. The proposed strong proximity condition, along with the polynomial upper bound in efficiency given in Theorem 2 (with Algorithm 1) is the main positive result of the paper which hints towards the use of strong proximity-like measures as opposed to classical ones that simply assume similarity in rewards and transitions.

3. The paper is very well-written and excels in clarity, flow and intuitive motivations and explanations of each property, definition and result.


## Weaknesses

As stated in my summary, the only weaknesses of this paper lie in the assumptions made.

1. One of the assumptions is already mentioned as a limitation in the discussion section -- the assumption that the MDPs share a state-action space, which does not hold in most real-life applications.

2. The other weakness is that the lower bound is a “worst case” bound -- it is not often the case that the featurization of the state-action space completely hides the shared near-optimal policy for a family of MDPs. A more relevant (positive or negative) result would be something that relaxes this assumption i.e. allows for informative featurizations and studies similar proximity conditions for guaranteeing generalization efficiency.


**Time Spent Reviewing:**

4

---

> ### Author Response · Authors · 2021-08-09
> **Thank you**
>
> Thank you for your review and your kind words about our paper. Below, we address your questions. Please let us know if anything remains unclear.
>
> **1. One of the assumptions is already mentioned as a limitation in the discussion section -- the assumption that the MDPs share a state-action space, which does not hold in most real-life applications.**
>
> We agree that the assumption that the MDPs share a state-action space limits the applicability of our upper bound. However, the reason we studied the shared state-action space setting, is because we proved a lower bound for this setting in the case of Weak Proximity, so it is natural to study this same setting when considering Strong Proximity to make the results more comparable. Indeed, we highlight that the shared state-action space assumption *strengthens* our lower bound, since a lower bound which applies to this special case of course applies to the more general realistic case of when the MDPs have different state spaces. Finally, as we said in the Future Work subsection of our Discussion, we remark that if one is given a feature map which maps the state spaces of different MDPs to the same feature space, then our upper bound insights can conceivably be applied.
>
> **2. The other weakness is that the lower bound is a “worst case” bound -- it is not often the case that the featurization of the state-action space completely hides the shared near-optimal policy for a family of MDPs. A more relevant (positive or negative) result would be something that relaxes this assumption i.e. allows for informative featurizations and studies similar proximity conditions for guaranteeing generalization efficiency.**
>
> Thank you for raising this excellent point! We absolutely agree that our lower bound is worst case. Nevertheless, a few comments are in order.
>
> First, we emphasize that the featurization in our construction satisfies SIO. In particular, each MDP shares the same featurization, but for any individual MDP, that same featurization is expressive enough that it permits an optimal linear policy for that MDP. This is a non-trivial requirement, and ostensibly, a featurization that satisfies this requirement might be useful. Indeed, at least a priori, it is reasonable to expect that such a featurization might give some information about the true shared optimal policy (although we prove that in general it does not).
>
> Second, before we can study featurizations that are more informative, we must understand what sorts of featurizations are worst case in the first place. Indeed, a very long line of work in RL theory has studied such worst case featurizations (see, for instance, the papers [A], [B], [C]), where a featurization *appears* useful, but it is actually not. We view our work as taking an initial step towards understanding what sort of properties are necessary for a featurization to be useful for generalization. We prove (amongst other things) that even though all the MDPs have the same featurization that permits an optimal linear policy for each MDP, this featurization can still not be useful in general. But we absolutely believe that future work should build off this, and study other featurizations that are less worst case, and show efficient generalization guarantees. We are happy to mention this in the Discussion and Future Work section.
>
> **Papers Cited:**
>
> [A] Du, Kakade, Wang, Yang. Is a Good Representation Sufficient for Sample Efficient Reinforcement Learning? 2019.
>
> [B] Wang, Foster, Kakade. What are the Statistical Limits of Offline RL with Linear Function Approximation? 2021.
>
> [C] Weisz, Amortila, Szepesvári. Exponential Lower Bounds for Planning in MDPs With Linearly-Realizable Optimal Action-Value Functions, 2021.

---

### Official Review · Reviewer_yyWK · 2021-07-19

**Rating:** 7
**Confidence:** 4

**Summary:**

This paper studies conditions on which generalization across tasks in RL is feasible. The authors introduce the concept of Weak Proximity to capture problems where MDPs are similar and each can be easily solved (for the initial state). They show despite such a nice assumption, to enable generalization over the MDP distribution or fast adaption requires exponential complexity (wrt horizon) in general. They further define a Strong Proximity assumption, which enables generalization, though being less practical.

**Limitations And Societal Impact:**

Yes, although the wording about the condition for the upper bound can be more precise.

**Main Review:**

Originality: The contribution here is original to my knowledge. The insights here are quite interesting, though in the hindsight obvious (because the authors did a good job in writing the proof).

Significance: Generalization is an important topic in RL. This paper provides basic principles under which generalization can be feasible in general. These insights would be valuable for future work.

Clarity: This paper is very well written, despite its theoretical nature. In addition, the proofs in the appendix are well organized as well.

Quality: I could not find major mistakes in the paper, but I admit I was not able to go through the full appendix. Here're some remarks.

1. The Average Performance setting is a not an MDP but a POMDP problem. I'm wondering if some existing results from POMDP would directly imply the learning difficulty here.

2. Consider an alternate mixture setting where the reward and transition in every time step is independently sampled from a family of MDPs. This is also a common trick people use to enable generalization in practice. Does the learning difficulty still apply? If the MDPs are linear, finding a good policy seems easy. But maybe this is not true with Weak Proximity assumption. Or generally, if the MDPs have Weak Proximity and are linear MDPs of a known feature, does the difficulty still persist?

3. The upper bound requires Strong Proximity "and" the MDPs sharing the same deterministic dynamics. However, in the introduction and many other places, it only mentions or only emphasizes the need of Strong Proximity. This can be confusing. Please highlight the two together or bring the assumption on dynamics as part of the Strong Proximity assumption, especially where the Limitation is mentioned.

4. If the optimization oracle can efficiently return all the optimal policies of an MDP, then it seems the difficulty in the counterexample no longer applies? Though this is not provided by the Weak Proximity assumption.


**Time Spent Reviewing:**

4

---

> ### Author Response · Authors · 2021-08-09
> **Thank you**
>
> Thank you for your review and your kind words about our paper. Below, we address your questions. Please let us know if anything remains unclear.
>
> **1. "The Average Performance setting is a not an MDP but a POMDP problem. I'm wondering if some existing results from POMDP would directly imply the learning difficulty here."**
>
> We do not believe that existing results from the POMDP literature imply the learning difficulty here. In many practical settings, a natural way to cast the Average Performance setting as a POMDP problem is as follows. Assume that there is a base MDP $M$, and a parameterized function $\phi_{\theta}$ that maps $M$ to a new MDP $M_{\theta}$ for each value of $\theta$. Different $M_{\theta}$ have different rewards and transitions, and they support the distribution $\mathcal{D}$. What makes this setting a POMDP, is that instead of seeing its "invariant" true state, the agent receives a high dimensional observation, which encodes some information of its true state in $M$ that is independent of $\theta$, and some noise component that depends on $\theta$. This inherently makes the problem more difficult, since the agent has only partial observability of its true state, in addition to the fact that the rewards and transitions vary with $\theta$. This, for instance, is how the paper [C] frames the problem, and we have cited this paper in our Related Work section (although we note that their study is empirical rather theoretical).
>
> By contrast, we study a simpler setting, where all the environments share the same state space, and the agent always receives full information of the state it is in. Please note that this only makes our lower bounds stronger, since the agent always has more information. Furthermore, in addition to varying rewards/transitions as described above, our lower bound setting has a number of additional requirements --- such as the existence of an algorithm for efficient individual MDP optimization, the fact that a linear policy is optimal for any MDP, and that the agent has access to a generative model that allows it to transition to any state of its choice. We believe that such requirements are most naturally phrased in the language of MDPs. While it is perhaps possible to describe these in the language of POMDPs, it is not immediately clear to us how this is done --- for instance, what does it mean to say that an agent has access to a generative model that allows it to transit to any state, if the agent only has partial observability of its state? Even if such a reduction to POMDPs is possible, it is not clear to us whether such a reduction is useful for the proofs in any sense.
>
> In case you had a different idea of how to reduce the Average Performance setting to a POMDP, then we request that you please provide us with more details. This will allow us to answer your question in a more precise fashion.
>
> **2. (a) "Consider an alternate mixture setting where the reward and transition in every time step is independently sampled from a family of MDPs. This is also a common trick people use to enable generalization in practice. Does the learning difficulty still apply?"**
>
> Yes, the learning difficulty certainly applies in this setting! To see this, note that for our lower bound we work with the SQM oracle model, where the agent has access to a generative model, and also incurs zero cost for sampling new MDPs from $\mathcal{D}$. Using this oracle model, the agent can easily run an algorithm of the sort you have described. In particular, say the agent is running some algorithm and it is currently at state $s$ in MDP $M$. If the agent wants to sample a transition/reward independently from the family, then it can simply sample a new MDP $M' \sim \mathcal{D}$ with zero cost, and then use the generative model to transition to $s$ in $M'$ with unit cost. Then it can directly take an action from $s$ in $M'$ and observe the reward/transition, which corresponds to sampling the reward/transition independently from the family. So our lower bound certainly applies to this sort of algorithm.
>
> **2. (b) "If the MDPs are linear, finding a good policy seems easy. But maybe this is not true with Weak Proximity assumption. Or generally, if the MDPs have Weak Proximity and are linear MDPs of a known feature, does the difficulty still persist?"**
>
> We understand your question as asking about linear MDPs as defined in the paper [A]. It is of course true that for an individual linear MDP, finding an optimal policy is easy. But for the case when each MDP supporting $\mathcal{D}$ is linear (possibly with the additional Weak Proximity condition), we are not aware of whether generalization is tractable or not. Our paper does not study linear MDPs, and so answering this question satisfactorily would require non-trivial further investigation. Note that although our SIO property requires MDPs to have an *optimal linear policy*, this is weaker than when all the MDPs are linear MDPs. We would also like to mention that one of the reasons we did not study linear MDPs is because this class is known to be *extremely* restricted (see, for instance, the discussion in the paper [B]).
>
> **3. "The upper bound requires Strong Proximity "and" the MDPs sharing the same deterministic dynamics. However, in the introduction and many other places, it only mentions or only emphasizes the need of Strong Proximity. This can be confusing. Please highlight the two together or bring the assumption on dynamics as part of the Strong Proximity assumption, especially where the Limitation is mentioned."**
>
> Sure, we absolutely agree that we should highlight Strong Proximity and shared deterministic dynamics together. We will highlight them together in the camera ready version, and will also emphasize it more in the Limitations section.
>
> **4. "If the optimization oracle can efficiently return all the optimal policies of an MDP, then it seems the difficulty in the counterexample no longer applies? Though this is not provided by the Weak Proximity assumption."**
>
> It is true that if an optimization oracle could return all the optimal policies of an MDP, then generalization would be easy (at least in our lower bound construction). However, typical RL methods optimize for a single policy with high value when deployed in an individual MDP, and achieving this alone is often non-trivial. Hence we are not aware of any means to (query efficiently) implement such an oracle, and don't believe such an oracle is reasonable. Note also that in our lower bound construction, each MDP has $\mathcal{O}(2^H)$ different policies achieving optimal value. So any oracle which returns all optimal policies would immediately also be computationally inefficient in the context of our counterexamples.
>
> **Cited Papers:**
>
> [A] Jin, Yang, Wang, Jordan. Provably Efficient Reinforcement Learning with Linear Function Approximation, 2019.
>
> [B] Malik, Pacchiano, Srinivasan, Li. Sample Efficient Reinforcement Learning In Continuous State Spaces: A Perspective Beyond Linearity, 2021.
>
> [C] Song, Jiang, Tu, Du, Neyshabur. Observational Overfitting In Reinforcement Learning, 2019.

---

> > ### Comment · Reviewer_yyWK · 2021-08-31
> > **RE: Thank you**
> >
> > Thanks for the detailed responses. It is a good paper overall. Below I clarify some comments I made.
> >
> > I see the lower bound is tighter here. Even though I think the setting here is a pomdp, I agree with you that the general pomdp language likely doesn't have a compact way to describe the setting here.
> >
> > For the alternate mixture setting, I was trying to describe a different problem setting rather than an algorithmic approach to the current setting. I asked, because I was curious whether the difficulty here arises due to partial observability. The alternate mixture setting is potentially an easier setting. I agree with you that such an algorithmic approach cannot break the lower bound here.
> >
> > Regarding to "returning all possible optimal policies", yes, I agree in general that is difficult. But you seem to point out also that if the optimal policy is unique, then the lower bound construction needs to be changed perhaps? But maybe in this case the gap would depend on the gap to the optimal value, though I see this is beyond the scope here.

---

> > > ### Author Response · Authors · 2021-09-01
> > > **Re**
> > >
> > > **Re: Alternate Mixture**
> > > Since this is a different problem setting, it is not immediately clear to us whether the learning difficulty will apply, and this would merit future investigation.
> > >
> > > **Re: Unique Optimal Policy**
> > > Indeed, if the optimal policy is unique and shared by all the MDPs, and optimizing for the optimal policy in any individual MDP is easy (as guaranteed by SIO), then generalization becomes easy and our lower bound does not apply. Critical to our lower bound construction is the fact that many different optimal policies.
> > >
> > > Our lower bound can reasonably extend to a setting where there is a unique optimal policy, but only when there are many different *near*-optimal policies. In this scenario, if we want a lower bound, then the suboptimality of the SIO would necessarily have to be strictly larger than zero. Indeed, as you correctly mention, the suboptimality of the SIO would precisely scale as the gap between the value of the *near*-optimal policies (returned by SIO in any individual MDP), and the value of the (unique) optimal policy. This is in contrast to the suboptimality of SIO in our current lower bound, which is exactly zero (i.e. no suboptimality).
> > >
> > >
> > > Please let us know if there are any other questions or comments.

---

### Author Response · Authors · 2021-08-09
**General Response To All Reviewers - Thank You!**

We thank the reviewers for making time to read our paper, and for their very positive comments about our work. We are especially grateful that the reviewers were interested in our proofs.

Each reviewer has asked questions, and raised points that we believe will help improve the quality of the final camera ready version. We have responded to each reviewer's questions individually. If the reviewers have additional questions, then please do not hesitate to post them, and we will respond to them ASAP during the discussion phase.

---

### Decision · Program_Chairs · 2021-09-27

**Decision:**

Accept (Poster)

**Comment:**

The paper studies the important problem of generalization in RL. The authors proposed different metrics of similarity between MDPs and derive negative or positive results depending on the strength of the assumption. Based on reviews and rebuttal, I believe this a solid paper with interesting and non trivial results. Some assumptions seem rather strong but I'm persuaded that the paper is an interesting starting point for further research on the topic. So I'm proposing acceptance for it.